# SURGE: SURPRISE-GUIDED TOKEN REDUCTION FOR EFFICIENT VIDEO UNDERSTANDING WITH VLMS

**Chong Tang**[*]
University of Southampton
UCL AI Centre
Southampton, United Kingdom
chong.tang@soton.ac.uk

**Sannara Ek**
University of Cambridge
Cambridge, United Kingdom
se521@cam.ac.uk

**Dirk Koch**
University of Manchester
Manchester, United Kingdom
dirk.koch@manchester.ac.uk

**Robert Mullins**
University of Cambridge
Cambridge, United Kingdom
robert.mullins@cl.cam.ac.uk

**Alex Weddell**
University of Southampton
Southampton, United Kingdom
asw@ecs.soton.ac.uk

**Jagmohan Chauhan**
University College London
UCL AI Centre
London, United Kingdom
jagmohan.chauhan@ucl.ac.uk

## ABSTRACT

Videos contain rich information but also high redundancy, as consecutive frames often share similar backgrounds and predictable motions. Current video-language models (VLMs) are unable to exploit this redundancy and therefore perform a significant amount of superfluous computation, processing thousands of patch tokens even when little new information is present. What is missing is an on-the-fly, model-agnostic signal of temporal predictability to decide whether tokens carry unpredictable information that merits computation. We propose SURGE, a training-free and backbone-agnostic method that measures surprise in token space. Surprise scores are defined by the prediction error of each token from its recent history; high-surprise tokens are retained, while predictable ones are pruned. Aggregating scores over time produces a surprise curve that highlights key events, which can be further refined with CLIP-based query relevance to form a compact spatio-temporal mask. Experiments on multiple video understanding benchmarks show that SURGE reduces tokens by up to $7\times$ and prefill cost by **86–98%**, while maintaining accuracy within $\pm\mathbf{1}$ point of full-token baselines. By aligning computation with novelty, SURGE enables video VLMs to handle long contexts efficiently and without retraining. https://github.com/BarryTang22/SURGE.git

## 1 INTRODUCTION

VLMs face a fundamental scalability problem in video understanding tasks. Even short clips expand into thousands of visual tokens, and longer videos quickly overwhelm memory and computational resources. The quadratic complexity of attention makes long-context reasoning especially expensive, and practical deployments add further constraints such as streaming inputs, limited hardware, and the inability to retrain models. Recent large-scale systems, including InternVL (Chen et al., 2024b), Qwen series VL (Wang et al., 2024a; Bai et al., 2025), and LLaVA-Next (Li et al., 2024a), demonstrate strong video understanding but expose the steep computational price for long inputs.

---

[*]corresponding author

To address this bottleneck, research has explored efficiency along temporal, representational and token-level directions. Temporal methods reduce cost by selecting a subset of frames (Hu et al., 2025; Park et al., 2024), which can be effective for long-video QA but may miss fine-grained events within frames. Representation-focused approaches instead compress each frame into fewer tokens, for example by distilling features into compact query embeddings or by learning hierarchical representations that adjust granularity to budget (Li et al., 2023; 2025). At the token level, merging methods collapse redundant patches across space and time (Bolya et al., 2023; Lee et al., 2024; Choudhury et al., 2024), while pruning methods remove less useful tokens during inference (Zhang et al., 2024). These strategies are effective, yet many introduce complexity by requiring auxiliary selectors, fine-tuning of the backbone, or access to internal attention maps that are not always available in deployed systems. What is still missing is a training-free, backbone-agnostic criterion that can allocate computation more flexibly as a video unfolds.

A natural way to think about such a signal is to ask: **which parts of a video actually *surprise* the model as it evolves over time?** We define surprise as the prediction error of a token given its recent history: low error indicates predictable, redundant content, while high error signals novel events that merit computation. In videos, where continuity is common, surprise provides a clear criterion for allocating effort. Based on this idea, we introduce SURGE, a training-free and backbone-agnostic surprise mask that directs computation to new content. After the vision encoder produces patch tokens for each frame, SURGE applies a lightweight temporal predictor to estimate each token from its recent history. The surprise score is the difference between the estimate and the actual embedding: high-surprise tokens are kept, while predictable ones are pruned. Aggregating token scores over time yields a surprise curve which, after smoothing and peak picking, provides key event windows. The surprise mask is usable as is for efficient inference, and can be further refined with CLIP-based query relevance: we reweight event windows by query–keyframe similarity so the final spatio-temporal mask prioritizes content that is both new and on-topic. This reduces tokens before the language interface, adds negligible overhead, and fits naturally with other methods (e.g. AKS (Tang et al., 2025)). Across comprehensive experimental analysis, SURGE reduces tokens by up to $7\times$ and prefill cost by 86–98%, while keeping accuracy within $\pm 1$ point of full-token baselines. The pipeline of SURGE is shown in Fig. 2. Our contributions can be summarized as:

- A training-free method that measures temporal predictability in token space, yielding surprise curves and token masks that work with any ViT-based backbone without retraining.

- A compact spatio-temporal masking scheme, effective on its own and further refinable with CLIP for query-aware event selection.

- Extensive experiments across models, benchmarks and pruning budgets, demonstrating SURGE's robustness and its ability to combine with complementary strategies (e.g., keyframe selection) for additional gains.

## 2 RELATED WORK

**Sparsity as Selection.** A common strategy for VLM efficiency is to process only a sparse subset of the visual stream, either across time or at the token level. Temporal sparsity assumes most frames are redundant, so a small set of keyframes suffices. Methods such as SeViLA (Li et al., 2023; Yu et al., 2023), ViLA (Wang et al., 2024b), and AKS (Tang et al., 2025) follow this idea, improving long-video QA but sometimes missing fine-grained evidence when relevance estimates are imperfect (Park et al., 2024; Hu et al., 2025). Recent work such as BOLT (Liu et al., 2025) and ViLAMP (Cheng et al., 2025) extend this by jointly selecting relevant frames and condensing information hierarchically. Token sparsity instead prunes less useful patches within frames. Approaches like FastV (Shu et al., 2025), IVTP (Huang et al., 2024), ATP-LLaVA (Ye et al., 2025), and SparseVLM (Zhang et al., 2024) typically rely on attention thresholds or relevance scores. More recent methods like DyCoke (Tao et al., 2025) and DivPrune (Alvar et al., 2025) aim to dynamically retain diverse or cache-critical tokens, achieving strong accuracy without retraining. These methods are effective, yet often need careful calibration of thresholds and policies, and some depend on internal attention maps often unavailable in deployed systems (Wei et al., 2023; Chen et al., 2024a). Realistically, sparsity proxies highlight what seems important at the current step but not what has changed, so redundant but on-topic segments may still absorb computation while unseen events can be overlooked.

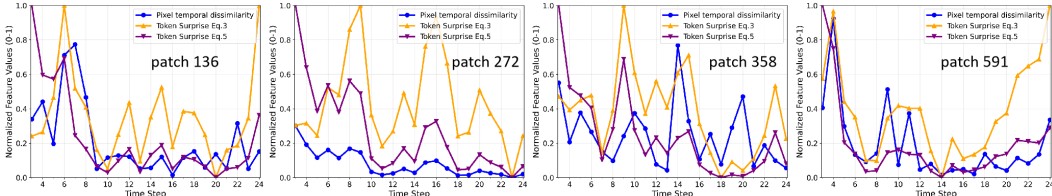

Figure 1: Token surprise vs. pixel change. For four random patches, we compare pixel dissimilarity (blue), raw surprise (orange; Eq. 3), and normalized surprise (purple; Eq. 5). Normalization suppresses drift and aligns peaks with true content changes.

**Representation Compression.** Another strategy is to compress visual representations so fewer tokens are processed per frame or interval. Some methods replace dense patches with compact learned embeddings: LLaMA-VID encodes each frame into just two tokens via cross-attention (Li et al., 2024b), Video-XL learns adaptive summarization tokens (Shu et al., 2025), and Matryoshka generates nested coarse-to-fine token sets for dynamic accuracy–efficiency trade-offs (Cai et al., 2024). These designs achieve large tokens reductions but require retraining, architectural changes, or additional hyperparameters. Another complementary approach is token merging (ToMe) (Bolya et al., 2023), which progressively merges similar patches during inference to reduce redundancy. Extensions to video include methods such as vid-TLDR (Choi et al., 2024) and TempMe (Shen et al., 2024a), as well as spatio-temporal schemes like STTM (Feng et al., 2024; Lee et al., 2024; Choudhury et al., 2024). Recent works like VisionZip Yang et al. (2025), LongVU (Shen et al., 2024b) and Chat-UniVi (Jin et al., 2024) further combine spatial and temporal compression to enable long-horizon video processing. Two-stage pipelines like PruMerge (Shang et al., 2024) combine pruning and clustering for up to $14\times$ reductions. While effective, these approaches depend on architectural modifications, merge schedules, or retrained summarizers that are often model- or dataset-specific.

**Predictability and Surprise.** Beyond sparsity and compression, a longstanding idea in cognitive science and machine learning is to use prediction error (*surprise*) as a signal of information value. Predictive coding suggests that expected inputs are suppressed while unexpected ones receive deeper processing (Itti & Baldi, 2005), and curiosity modules in reinforcement learning reward high prediction error to drive exploration (Pathak et al., 2017). In video, frame-level errors have been used for anomaly detection (Liu et al., 2018). The common principle is that novelty, defined as a "deviation from expectation", deserves computation. By contrast, most VLM efficiency methods rely on proxies such as attention weights, similarity scores, or trained selectors (Zhang et al., 2024; Huang et al., 2024; Ye et al., 2025), which add complexity and often depend on model-specific thresholds or attention maps that may not transfer across datasets or deployment settings.

We argue that a more intuitive criterion is to test predictability itself: tokens that align with prior context contribute little new information, while those that diverge from expectation indicate genuine change. Unlike attention weights or similarity scores, this signal directly measures novelty as it unfolds, offering a lightweight and training-free mechanism that complements sparsity and compression by aligning computation with new information.

## 3 FROM PREDICTABILITY TO SURPRISE IN TOKEN SPACE

### 3.1 PRELIMINARIES AND TOKEN DYNAMICS

Given a sequence of frames $I \in \mathbb{R}^{T \times C \times H \times W}$, the vision tower partitions each frame into $m$ spatial cells and produces patch embeddings $Z_t = [z_t^{(1)}, \ldots, z_t^{(m)}] \in \mathbb{R}^{m \times d}$, where $z_t^{(j)} \in \mathbb{R}^d$ is the token for spatial index $j$. Formally, $z_t^{(j)} = f_j(I_t)$, where $f_j$ is the encoder map for cell $j$. Natural videos evolve smoothly: consecutive frames satisfy $I_{t+1} \approx I_t + \Delta I_t$ with bounded perturbation $\Delta I_t$. Since $f_j$ is differentiable, a first-order Taylor expansion gives

$$z_{t+1}^{(j)} \approx z_t^{(j)} + J_{f_j}(I_t)\, \Delta I_t, \tag{1}$$

where $J_{f_j}(I_t)$ is the Jacobian of $f_j$ at $I_t$. This implies approximately linear dynamics in token space:

$$z_{t+1}^{(j)} - 2z_t^{(j)} + z_{t-1}^{(j)} \approx 0. \tag{2}$$

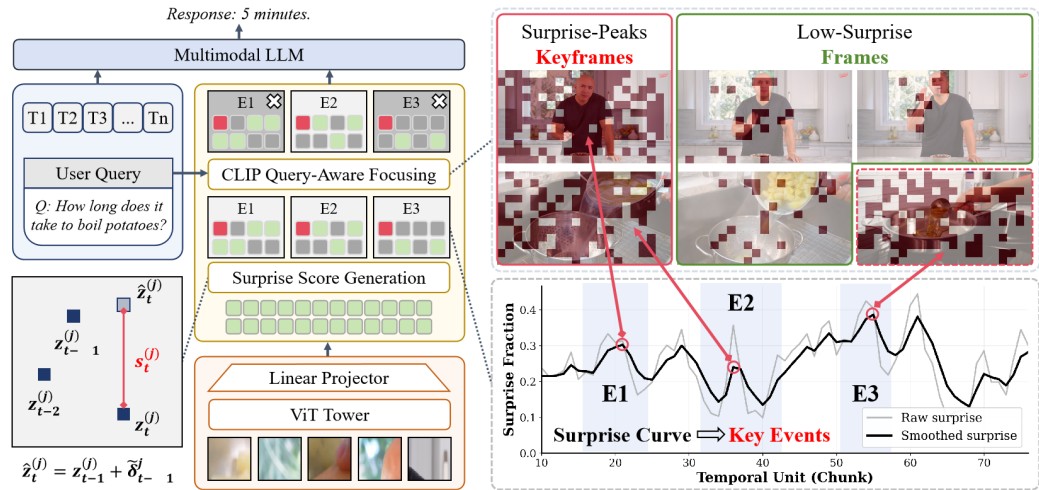

Figure 2: **SURGE pipeline.** Tokens are predicted with a constant-velocity model, then detrended and variance-normalized to yield surprise scores. This surprise computation is performed directly on the raw patch embeddings output by the vision tower before projection into the language model. We retain the top-$\rho$ tokens, aggregate them into a surprise curve to segment key events, and optionally rank events with CLIP (Top-$K$). Finally, only high-surprise tokens from key events are forwarded to the LLM, while others are pruned. Note, the SURGE block appears after the projection layer to reflect where the masking is applied, but the surprise scores themselves are computed earlier, on pre-projection features.

Eq. 2 encodes a constant-velocity prior: smooth motions satisfy it, while abrupt events yield large deviations interpreted as *surprise*. This connects to representation learning approaches that model videos as smooth stochastic processes (e.g., Brownian bridges) (Zhang et al., 2023) and to frame interpolation methods that rely on constant-velocity or higher-order motion priors (Zhong et al., 2024). We adopt the same principle at the token level, using deviations from constant-velocity prediction as surprise signals that highlight semantically meaningful events.

## 3.2 TRAINING-FREE TEMPORAL PREDICTION

Building on Sec. 3.1, we can use a causal constant-velocity predictor, common in tracking and interpolation, to estimate tokens from recent displacements.

**Constant-velocity extrapolation.** For token $j$, let $\Delta z_t^{(j)} = z_t^{(j)} - z_{t-1}^{(j)}$ denote its displacement between consecutive steps. A natural causal predictor is to carry forward the most recent displacement,

$$\hat{z}_t^{(j)} = z_{t-1}^{(j)} + \tilde{\delta}_{t-1}^{(j)}, \qquad \tilde{\delta}_{t-1}^{(j)} \approx z_{t-1}^{(j)} - z_{t-2}^{(j)}. \tag{3}$$

For $t \leq 2$, we initialize $\tilde{\delta}_1^{(j)} = 0$ and set $\hat{z}_2^{(j)} = z_1^{(j)}$. This autoregressive extrapolation is consistent with the constant-velocity prior in Eq. 2. It is causal, training-free, and applies uniformly across spatial indices, ensuring compatibility with any ViT backbone. However, this naive form cannot distinguish novelty from large-scale coherent motion, arising from camera panning, scene-level shifts, or other global transformations, which induce nearly uniform changes across tokens (Lian et al., 2023).

**Global drift correction.** Such drift is spatially smooth and can dominate residuals, causing false surprise (Lian et al., 2023; James et al., 2023). To suppress this effect, we approximate the displacement field with an affine function of spatial coordinates, a standard choice in motion compensation and stabilization (Nie et al., 2024). Let $(x_j, y_j)$ denote the normalized position of token $j$, and define

$$\Delta z_t^{(j)} \approx c_0 + c_x x_j + c_y y_j,$$

with coefficients $c_0, c_x, c_y \in \mathbb{R}^d$ fit by least squares. Concretely, let $X = [\mathbf{1}, x, y] \in \mathbb{R}^{m \times 3}$ stack token coordinates and $\Delta Z_t \in \mathbb{R}^{m \times d}$ stack displacements row-wise; the closed-form is

$\hat{C} = (X^\top X)^{-1} X^\top \Delta Z_t \in \mathbb{R}^{3 \times d}$, with rows corresponding to $c_0^\top, c_x^\top, c_y^\top$. The detrended displacement is then

$$\tilde{\delta}_t^{(j)} = \Delta z_t^{(j)} - (c_0 + c_x\, x_j + c_y\, y_j). \tag{4}$$

This removes global translation ($c_0$) and first-order planar flow ($c_x, c_y$), ensuring that residuals reflect true surprise rather than scene-level drift. Special tokens (e.g., *[CLS]* or register tokens) are excluded from this fit and always retained. The $3 \times 3$ least-squares solve per frame adds negligible $O(md)$ cost. Finally, Eq. 3 is applied using the detrended displacement $\tilde{\delta}_{t-1}^{(j)}$ to produce the causal prediction.

**Surprise scoring.** Given the causal prediction $\hat{z}_t^{(j)}$, we compute the *surprise vector*: $e_t^{(j)} = z_t^{(j)} - \hat{z}_t^{(j)}$, which measures the prediction error of token $j$. Under smooth motion, the vector remains small but new information makes them large. To obtain a scalar measure, we define the *surprise* score of token $j$ by normalizing the squared magnitude with a running estimate of its variance:

$$s_t^{(j)} = \frac{\|e_t^{(j)}\|_2^2}{\sigma_t^{2,(j)} + \varepsilon}, \tag{5}$$

where $\sigma_t^{2,(j)}$ is an exponential moving average of past vector variances for token $j$, and $\varepsilon > 0$ prevents division by zero. This variance-normalized formulation provides a calibrated statistic: under locally linear–Gaussian dynamics with no change, $s_t^{(j)}$ stays near its expectation, while significant deviations directly indicate surprise.

As a sanity check, Fig. 1 shows that token-space surprise tracks pixel change, validating predictor residuals as a proxy. Specifically, the raw constant-velocity predictor (Eq. 3) often fires spuriously under global shifts such as camera pans. After applying drift detrending (Eq. 4) and variance normalization (Eq. 5), surprise spikes align more closely with true content changes, suppressing noise and yielding a stable, patch-consistent signal.

## 4    From Token Surprise to a Spatio-Temporal Mask

Given per-token surprise scores $s_t^{(j)}$ (Sec. 3), we produce a compact spatio-temporal mask in two stages: (i) global percentile thresholding to keep the most surprising tokens across the sequence, and (ii) construction of a *surprise curve* per temporal unit to segment key events. For query-focused applications, we then rank these events using CLIP similarity (Radford et al., 2021) at peak frames and concentrate computation on the top-$K$ relevant ones.

### 4.1    Adaptive token masking

Within each sequence, we adaptively keep the most surprising tokens according to a global percentile. Let $\mathcal{B}$ denote the current buffer of tokens (the full clip offline, or the observed prefix in streaming), and collect all surprise scores $\mathcal{S}_\mathcal{B} = \{ s_u^{(j)} : (u, j) \in \mathcal{B} \}$. The global $p$-percentile is $q(p) = \text{Quantile}_p(\mathcal{S}_\mathcal{B})$ for $p \in (0, 1]$. The binary mask is then

$$M_{u,j} = \mathbb{1}\left\{ s_u^{(j)} \geq q(p) \right\}. \tag{6}$$

This global selection retains the top $\rho = (1 - p)$ fraction where content changes: dynamic frames contribute more, redundant frames contribute less. Note that special tokens (e.g. *[CLS]*) are always preserved. The mask is applied after the vision encoder and positional encodings, and before the language model, making the procedure training-free, backbone-agnostic, and compatible with standard ViT-based pipelines.

### 4.2    Surprise curve and key events

After generating token-level masks, we aggregate surprise over time to detect salient events. For each temporal unit $u$ (a frame or a chunk), we count how many tokens exceed the global threshold: $S_u = \sum_{j=1}^m M_{u,j}$, equivalently the fraction $S_u/m$. The sequence $\{S_u\}$ forms the *surprise curve*. We smooth it with an exponential moving average, $\bar{S}_u = \gamma\, \bar{S}_{u-1} + (1 - \gamma)S_u$, to reduce noise.

**Peak-based event segmentation.** Let $\mathcal{P} = \{\tau_1 < \cdots < \tau_n\}$ be the local maxima (peaks) of $\bar{S}_u$, found with minimum separation $\Delta$ and a small prominence threshold. Let $U$ be the total number of units. We define event boundaries at midpoints between adjacent peaks:

$$b_0 = 1, \qquad b_k = \left\lfloor \frac{\tau_k + \tau_{k+1}}{2} \right\rfloor \ (k = 1, \ldots, n-1), \qquad b_n = U, \tag{7}$$

and assign each peak $\tau_k$ the event interval $\mathcal{I}_k = [\, b_{k-1}, \, b_k \,)$, $k = 1, \ldots, n$. Thus, *durations between peaks* are taken as key events.

**Query-aware event focusing.** To align events with a text query $q$, we use CLIP similarity at the peak frames. Let $v_{\tau_k}$ be the frame-level embedding at peak $\tau_k$ and define $r_k = \text{sim}(q, v_{\tau_k})$. We rank events by $\{r_k\}$ and keep the top-$K$: $\mathcal{E}_K = \text{TopK}_k(r_k)$. We then concentrate computation on $\{\mathcal{I}_k : k \in \mathcal{E}_K\}$ by applying $M_{u,j}$ within these intervals, while maintaining only a small context floor elsewhere. Although this adds one CLIP pass over the set of peak frames, it further reduces tokens forwarded to the language model and improves focus in very long-context retrieval. Finally, based on the surprise mask $M_{u,j}$, event intervals $\{\mathcal{I}_k\}$, and CLIP-selected indices $\mathcal{E}_K$, define the event indicator $A_u = \mathbb{1}\{\, u \in \bigcup_{k \in \mathcal{E}_K} \mathcal{I}_k \,\}$. With a small context floor $C_{u,j}$ (e.g., the top-$k_{\text{ctx}}$ tokens by $s_u^{(j)}$ when $A_u = 0$), the final mask is (see Appendix A.4 and A.5 for additional visualizations):

$$M_{u,j}^\star = A_u \cdot M_{u,j} + (1 - A_u) \cdot C_{u,j}. \tag{8}$$

## 5 Experiments

**Benchmarks & Baselines.** We evaluate SURGE on five representative video–language benchmarks: (1) **Video-MME** (Fu et al., 2025): overall QA on short–long clips; (2) **MLVU** (Zhou et al., 2025): long-video multi-task evaluation (M-Avg/G-Avg) including Needle QA, grounding, and summarization; (3) **MMBench-Video** (Fang et al., 2024): curated multi-step QA probing compositional/temporal reasoning; (4) **TempCompass** (Liu et al., 2024): structured temporal QA on short clips; (5) **LongVideoBench** (Wu et al., 2024): very long videos emphasizing retrieval and cross-event reasoning. As baselines, we compare against random token pruning, **FastV** (token pruning/aggregation) (Shu et al., 2025) and **AKS** (adaptive keyframe selection) (Tang et al., 2025), chosen because they are publicly available, replicable, and applicable across model families.[1] We also report *AKS w/ SURGE* and *AKS w/ SURGE★* to assess complementarity with temporal selection. Full details and justifications are in Appendix A.3 and Appendix A.2.

**Models.** To demonstrate SURGE's breadth, we evaluate three flagship VLMs: (i) **InternVL-3.5-VL** (Wang et al., 2025) (8B; *15k-token* context): the latest open-source model with strong general-purpose video QA and extended text window; (ii) **Video-LLaVA-Qwen** (Lin et al., 2023) (7B; *64-frame* cap): a mid-size video variant with a strict frame budget, highlighting token efficiency; (iii) **Qwen2.5-VL** (Bai et al., 2025) (7B; *131k-token* context): a long-context model able to process hours-long videos, ideal for testing SURGE's scalability. Together these models provide a comprehensive generality test. Further implementation details and justifications are in Appendix A.1 and Appendix A.3.

**Evaluation pipeline & Hyperparameters.** We use **VLMEvalKit** (Duan et al., 2024) for benchmarks, applying SURGE purely at inference. Experiments run on 1–8 A100 80GB GPUs. By default, we retain the top $\rho = 0.25$ tokens (75th percentile), smooth surprise curves with EMA ($\gamma = 0.9$), and enforce a minimum separation $\Delta = 8$ units between peaks. For query-focused retrieval, we select the *Top-5* events by CLIP (ViT-B/32) similarity at peak frames. No context floor (Eq. 8) is used in the main experiments, isolating surprise-driven gains.

### 5.1 Performance

**Large token reductions with stable accuracy.** At the default $\rho=0.25$, SURGE retains only $\sim$26%–27% of visual tokens ($\sim$4$\times$ fewer), and SURGE★ with CLIP Top-5 further reduces this to $\sim$14%–16% ($\sim$7$\times$ fewer). Yet benchmark scores remain within $\pm 1$ point of the baseline (Tab. 1), validating our hypothesis that predictability in token space is a reliable proxy for redundancy

---

[1] Several other methods currently support only specific VLMs. For FastV, we follow its official implementation and adapt to Qwen2 LM by replacing the k-1 attention layer with an eager version to access attentions.

*Q: Who are the guards questioning in the facility?*

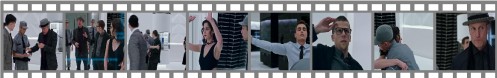

**Base Model**: *The guards are questioning a man in a suit and a woman in a black dress.*

**SURGE**: *The guards are questioning a group of people, including a man in a suit, who appears to be a magician or performer, and a woman in a black dress.*

**SURGE***: *The guards are questioning a group of people, including a man in a suit and a woman in a black dress. The man in the suit is seen gesturing and explaining something, while the woman in the black dress is also part of the conversation.*

**AKS**: *The guards are questioning a man in a suit and a woman in a black dress. The man is identified as the magician, and the woman is his assistant.*

**AKS w/SURGE**: *The guards are questioning a group of people in the facility. The scene shows a woman in a black dress and a man in a suit being questioned by the guards.*

**Random**: *The woman appears to be the focus of the questioning, as she is being held by the man and is the one being directly addressed by the guards.*

*Q: Which animals attack the explorers on the bridges?*

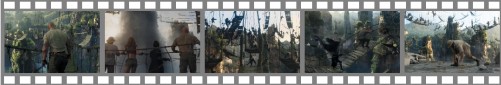

**Base Model**: *The explorers are attacked by baboons on the bridges.*

**SURGE**: *The animals that attack the explorers on the bridges are baboons.*

**SURGE***: *The animals that attack the explorers on the bridges are baboons.*

**AKS**: *The animals are baboons.*

**AKS w/SURGE**: *They are attacted by baboons.*

**Random**: *Gorillas.*

Figure 3: Qualitative examples. SURGE and SURGE$^\star$ can even provide more accurate and detailed answers than the base model.

Table 1: **Benchmark results (64 frames).** SURGE retains a global top-$\rho$=0.25 of tokens; SURGE$^\star$ adds CLIP Top-5 event focusing. "Tokens" reports average visual+fixed text tokens per sample at 64 frames. [†] Exceeds InternVL-3.5-VL's 15k-token context; inputs were truncated by the runtime, so scores reflect truncated context. Best performance is in **bold**.

| Model | Tokens | V-MME | MLVU (M / G) | MMB-V | T-Compass | LVB |
|---|---|---|---|---|---|---|
| InternVL-3.5-VL (8B) | 17,124[†] | 66.0 | 71.7 / 3.44 | 1.54 | 68.9 | 61.3 |
| + SURGE | 4,674 | 64.9 | 71.5 / 3.45 | **1.57** | **69.0** | 61.7 |
| + SURGE$^\star$ | 2,932 | 65.8 | 71.7 / **3.69** | **1.57** | 69.7 | **62.2** |
| + AKS | 17,124 | 66.1 | 71.6 / 3.48 | 1.54 | 68.6 | **62.2** |
| + AKS w/ SURGE | 4,819 | 65.9 | 71.7 / 3.39 | 1.56 | 68.6 | 62.0 |
| + AKS w/ SURGE$^\star$ | 2,390 | **66.2** | **72.3** / 3.58 | 1.55 | 68.4 | 61.9 |
| Video-LLaVA-Qwen (7B) | 12,246 | 63.4 | **72.9** / 3.30 | 1.53 | 66.9 | 58.3 |
| + SURGE | 3,324 | 63.1 | **72.9** / 3.30 | 1.53 | **67.0** | 59.1 |
| + SURGE$^\star$ | 1,884 | 64.5 | 72.7 / 3.20 | 1.60 | 66.9 | 61.9 |
| + FastV | 3,300 | 58.1 | 52.3 / 3.11 | 1.29 | 61.7 | 55.4 |
| + AKS | 12,246 | 64.7 | 72.7 / 3.30 | 1.55 | 66.9 | 61.4 |
| + AKS w/ SURGE | 3,157 | **64.9** | 72.4 / 3.27 | 1.55 | 66.7 | **62.3** |
| + AKS w/ SURGE$^\star$ | 1,961 | 62.4 | 72.8 / **3.56** | **1.61** | 66.9 | 62.2 |
| Qwen2.5-VL (7B) | 41,590 | 62.2 | 65.8 / **4.26** | 1.60 | 70.5 | 60.0 |
| + SURGE | 10,992 | 60.9 | 65.7 / **4.26** | 1.72 | 70.5 | 59.4 |
| + SURGE$^\star$ | 5,207 | **62.7** | **66.1** / 4.24 | 1.70 | 67.7 | **61.3** |
| + AKS | 41,590 | 62.0 | 65.8 / 4.26 | 1.67 | 70.7 | 59.9 |
| + AKS w/ SURGE | 11,588 | 61.9 | 65.8 / 4.25 | 1.68 | **71.1** | 60.3 |
| + AKS w/ SURGE$^\star$ | 6,204 | **62.7** | 66.0 / **4.26** | **1.73** | 70.9 | 60.4 |

(Sec. 3). The qualitative examples (Fig. 3) further illustrate this: even when the base model gives incomplete answers, SURGE and SURGE$^\star$ recover more accurate and detailed responses by focusing on novel and relevant tokens. In contrast, random pruning often drops critical evidence, producing unstable outputs, while surprise-driven selection preserves informativeness.

**Performance across contexts.** On benchmarks stressing long videos and cross-event reasoning, SURGE$^\star$ not only preserves but *surpasses* full-token baselines: e.g., in Tab. 1, InternVL-3.5-VL on **LVB** (+0.9) and **MLVU** G-Avg (+0.25), with consistent improvements on **T-Compass** (+0.8) and **MMB-V** (+0.03). These are precisely the conditions where surprise-based reallocation and query-aware focusing are most useful, confirming that SURGE effectively identifies and prioritizes new, relevant content (Sec. 4). For short-form QA benchmarks such as **V-MME**, where redundancy is lower, SURGE induces only small fluctuations (typically within ±1 point). Importantly, SURGE$^\star$ often recovers or slightly improves over the baseline (e.g., +1.1 on Video-LLaVA-Qwen), underscoring that SURGE is safe to apply even in short-video settings with relatively low redundancy.

**Comparison to pruning and keyframe baselines.** Against FastV (attention-based pruning), SURGE is markedly more reliable at the same budgets, e.g., on Video-LLaVA-Qwen, V-MME 64.5

Table 2: **Token-accuracy trade-off on Qwen2.5-VL.** $\rho$ is the fraction of visual tokens retained (1.0=full tokens). "$\pm\%$" indicates the maximum relative deviation (5 runs), reflecting result stability.

| Method | $\rho$ | V-MME | MLVU (M/G) | MMB-V | T-Compass | LVB |
|---|---|---|---|---|---|---|
| Qwen2.5-VL (7B) | 1.00 | 62.2 | 65.8 / 4.26 | 1.60 | 70.5 | 60.0 |
| + SURGE $\pm0.8\%$ | 0.75 | **62.3** | 65.8 / 4.22 | 1.60 | **70.3** | **60.5** |
| + SURGE$^\star$ $\pm1.1\%$ | 0.75 | 62.1 | **66.4** / 4.25 | 1.60 | 70.2 | 58.1 |
| + Random $\pm3.7\%$ | 0.75 | 61.9 | 65.7 / **4.26** | **1.61** | 69.7 | 50.1 |
| + SURGE $\pm0.7\%$ | 0.50 | 62.0 | 65.8 / **4.26** | 1.66 | **72.3** | **59.7** |
| + SURGE$^\star$ $\pm0.5\%$ | 0.50 | **62.1** | **66.2** / 4.22 | **1.70** | 71.7 | 57.9 |
| + Random $\pm4.2\%$ | 0.50 | 62.0 | 64.9 / 4.24 | 1.62 | 66.1 | 57.0 |
| + SURGE $\pm0.4\%$ | 0.25 | 60.9 | 65.7 / **4.26** | **1.72** | **70.5** | **59.4** |
| + SURGE$^\star$ $\pm0.6\%$ | 0.25 | **62.0** | **66.1** / 4.24 | 1.70 | 67.7 | 56.3 |
| + Random $\pm13.2\%$ | 0.25 | 53.6 | 55.9 / 3.32 | 0.96 | 57.4 | 49.7 |
| + SURGE $\pm0.7\%$ | 0.10 | **60.3** | **65.8** / **4.22** | **1.58** | **70.2** | **58.6** |
| + Random $\pm23.9\%$ | 0.10 | 36.8 | 37.4 / 3.11 | 0.94 | 50.9 | 46.0 |
| + SURGE $\pm1.0\%$ | 0.01 | **58.7** | **65.8** / **4.22** | **1.55** | **70.2** | **55.7** |
| + Random $\pm9.8\%$ | 0.01 | 29.8 | 35.0 / 3.10 | 0.84 | 50.6 | 37.3 |

Table 3: **Top-$K$ CLIP event focusing evaluation on Qwen2.5-VL (7B).** Baseline=full tokens, SURGE=percentile masking ($\rho$=0.25), SURGE$^\star$=SURGE ($\rho$=0.25) + CLIP Top-$K$ event selection.

| Method | $K$ | V-MME | MLVU (M/G) | MMB-V | T-Compass | LVB |
|---|---|---|---|---|---|---|
| Qwen2.5-VL (7B) | – | 62.2 | 65.8 / 4.26 | 1.60 | 70.5 | 60.0 |
| + SURGE | – | 60.9 | 65.7 / 4.26 | 1.72 | 70.5 | 59.4 |
| + SURGE$^\star$ | 1 | 51.7 | 37.9 / 2.51 | 1.09 | 23.9 | 40.1 |
| + SURGE$^\star$ | 3 | 59.0 | 60.5 / 3.70 | 1.15 | 59.7 | 49.6 |
| + SURGE$^\star$ | 5 | 62.0 | 66.1 / 4.24 | 1.70 | 67.7 | 56.3 |
| + SURGE$^\star$ | 7 | 62.3 | 66.8 / 4.30 | 1.65 | 70.5 | 60.2 |
| + SURGE$^\star$ | 10 | 62.7 | 66.8 / 4.26 | 1.71 | 71.0 | 60.2 |

vs. 58.1, MLVU 72.7/3.20 vs. 52.3/3.11 (Tab. 1), indicating that temporal surprise can be a stronger criterion than attention magnitude for deciding which visual tokens to keep in video understanding tasks. In contrast, AKS selects fewer frames instead of pruning tokens, and can match or surpass baselines on long-video metrics. Combined with SURGE, the two act complementarily, obtaining further gains in efficiency and accuracy, showing SURGE is both competitive alone and composable with temporal selection.

## 5.2 EFFICIENCY−ACCURACY TRADE-OFFS

**Token Efficiency Analysis.** At moderate pruning ($\rho$=0.50–0.75), SURGE maintains accuracy within $\pm1$ of baseline, while improves it for **T-Compass** (+1.8) and **MMB-V** (+0.1–0.2). This suggests that surprise-guided masking effectively discards predictable background while preserving transition-heavy evidence. CLIP focusing further helps in some cases (e.g., **MLVU**, +0.3 M-Avg), validating that query-aware peak selection reallocates budget toward relevant content.

The random pruning baseline provides an informative contrast: at moderate levels ($\rho\geq0.50$), it can approximate baseline accuracy, confirming that redundancy is indeed present in visual tokens. However, once more than 75% of tokens are dropped, performance becomes highly unstable, with maximum relative deviations exceeding 20% (Tab. 2), reflecting frequent loss of critical information. By comparison, SURGE stays within $\pm1.1\%$ across all pruning levels, remaining stable even under aggressive settings ($\rho$=0.10–0.01). This yields usable accuracy while exposing a clear compute–accuracy trade-off.

**Effect of Top-$K$ CLIP Focusing.** We investigate CLIP event focusing by varying $K$, the number of peak events retained (Table 3). Extremely small $K$ (e.g., $K$=1) collapses coverage and causes severe accuracy drops, while larger $K$ steadily restores performance: with $K$=5–10, SURGE$^\star$ of-

ten matches or exceeds the full-token baseline, especially on long-context benchmarks like **MLVU** and **LVB**. This illustrates the precision–coverage trade-off: small $K$ risks missing relevant events, whereas moderate $K$ balances query alignment with sufficient context. In practice, $K=5$–$7$ provides a robust setting, confirming that query-aware focusing complements surprise masking by allocating compute to both *new* and *relevant* content.

**Long-Context Performance.** On **MLVU** with **Qwen2.5-VL**[2], we test up to 3600 frames (Fig. 4). Accuracy rises for both baseline and SURGE up to 256 frames. Beyond ∼230 frames, the baseline exceeds A100 80GB VRAM, while SURGE/SURGE⋆ remain executable, extending to 1024 and 3600 frames. Performance drops when truncation dominates (e.g., 52.3/3.61 at 1024, 35.4/3.30 at 3600), but the key result is that SURGE raises the practical upper bound, enabling over an order-of-magnitude longer videos to be processed on fixed hardware, while maintaining near-baseline accuracy within the feasible range.

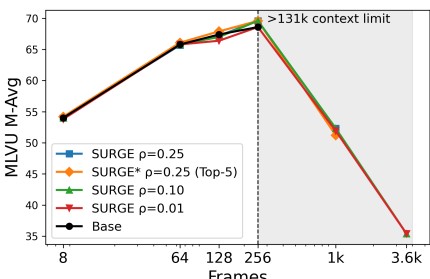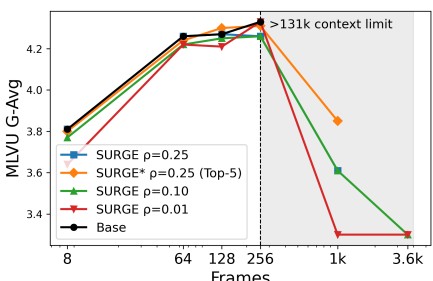

Figure 4: **Long-context evaluation on MLVU using Qwen2.5-V.** M-Avg and G-Avg across frame counts; shaded area exceeds the ∼131k-token limit. SURGE and SURGE⋆ extend capacity to 3600 frames while staying competitive below the limit.

**Efficiency Analysis.** On Video-MME with **Qwen2.5-VL (7B)**, SURGE achieves substantial prefill savings: at $\rho=0.25$, FLOPs/latency drop by **86%/79%** (Fig. 5(a),(c)), respectively, while generation FLOPs/latency also show modest reductions (**–38%/–14%** (Fig. 5(b),(d))). At extreme pruning ($\rho=0.01$), prefill costs shrink by over **98%**, and generation FLOPs are roughly halved. Total visual tokens reduce by **–72%** ($\rho=0.25$) to **–96%** ($\rho=0.01$). Together with accuracy stable at moderate pruning (Tab. 2), this shows SURGE translates token savings into substantial FLOP/latency gains, especially in the compute-heavy prefill stage. For SURGE⋆, a CLIP pass over 5–8 peaks adds roughly **0.63–1.0 TFLOPs/1027–1891 ms** per query, while pre-LLM pruning still removes most KV-cache and prefill load, keeping both variants effective under memory or throughput limits.

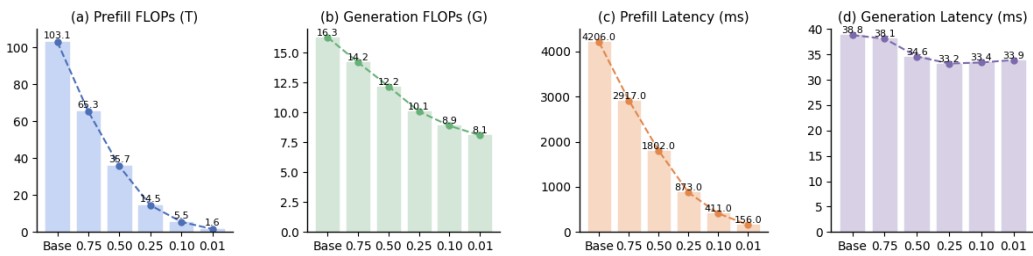

Figure 5: **Efficiency on Video-MME with Qwen2.5-VL (7B).** *X-axis* in all panels: token retention $\rho$. Approximate total token counts per setting: 41.6k, 31.6k, 21.5k, 11.6k, 5.5k, 1.8k.

## 5.3 COMPONENT ABLATIONS

Table 4 reports ablations on: **MLVU** (long multi-task), **TempCompass** (fine-grained temporal), and **MMB-V** (multi-step QA), chosen to cover complementary reasoning settings without redundancy. Removing drift detrend or variance normalization yields moderate but consistent drops, showing

---

[2]Extended ∼131k context window, ∼230 frames; beyond this inputs are truncated though outputs remain.

Table 4: **Component ablations at fixed budget.** Each row removes a single ingredient from SURGE (global affine drift detrending, variance normalization, or the causal temporal predictor), while keeping the global percentile mask and event segmentation unchanged.

| Variant (Qwen2.5-VL, $\rho$=0.25) | MLVU (M/G) | T-Compass | MMB-V |
|---|---|---|---|
| **SURGE** | 65.7 / 4.26 | 70.5 | 1.72 |
| w/o drift detrend (Eq. 4) | 64.9 / 4.18 | 69.4 | 1.65 |
| w/o variance norm (Eq. 5) | 65.1 / 4.22 | 69.7 | 1.70 |
| w/o temporal predictor (frame-diff only Eq. 3) | 63.4 / 4.17 | 66.9 | 1.55 |

their role in stabilizing surprise. The largest degradation comes from discarding the temporal predictor, where frame-differences mistake smooth motion for novelty. Overall, each component contributes to robustness. In addition, we also sweep hyperparameters and find SURGE to be stable: EMA smoothing $\gamma \in [0.7, 0.95]$, peak separation $\Delta \in [4, 12]$, $K$ in Tab. 3 and percentile in Tab. 2. With default settings, SURGE ($\gamma$=0.9, $\Delta$=8, $K = 5$, $\rho$=0.25) offers a good balance.

## 6 CONCLUSION

We introduced *SURGE*, a training-free and backbone-agnostic method that allocates computation to high-*surprise* content. SURGE addresses the scalability of video VLMs by cutting the cost of long inputs and expanding the *memory-bounded* effective context: by pruning predictable visual tokens before the multimodal LLMs, it reduces embedding/attention activations and KV-cache growth, which are typically constrained by the device VRAM/system memory and latency budgets. In practice, SURGE reduces token counts by up to $7\times$ and cuts prefill cost by nearly 90%, while keeping accuracy within $\pm 1$ point of full-token baselines. On commodity accelerators, this enables processing sequences that would otherwise hit out-of-memory (or unacceptable latency) for the same model; SURGE also composes with keyframe selection or query-aware focusing for added efficiency. A current limitation is that SURGE$^\star$ requires an extra CLIP pass and is sensitive to $K$ and query phrasing. Future work will explore lighter relevance models, adaptive event selection, and in-context alignment to improve robustness.

## ACKNOWLEDGMENTS

This work was supported by the Engineering and Physical Sciences Research Council (EPSRC), under the project *Perfect Recollection for Clearer Insight* (grant number EP/Y036077/1) and the computing resources and software from the NVIDIA Academic Grant program.

## REPRODUCIBILITY STATEMENT

We have taken several steps to ensure reproducibility. All models used in this work (InternVL-3.5-VL, Video-LLaVA-Qwen, and Qwen2.5-VL) are publicly available through HuggingFace, and Appendix A.1 details where SURGE is integrated into each codebase. Our implementation relies only on open-source tools and libraries, including HuggingFace Transformers, OpenAI CLIP, and VLMEvalKit for standardized evaluation on Video-MME, MLVU, MMBench-Video, Temp-Compass, and LongVideoBench. Hyperparameter ranges, default values, and ablation settings are reported in Section 5 and the appendix. Hardware (NVIDIA A100 80GB GPUs) and evaluation protocols (latency, FLOPs, token counts, and accuracy) are likewise fully specified. We will release our SURGE/SURGE$^\star$ modules, modified model wrappers, and experiment scripts upon acceptance to enable exact reproduction of all tables and figures. https://github.com/BarryTang22/SURGE.git

## ETHICS STATEMENT

This work focuses on improving the efficiency of video VLMs at inference. We do not collect new datasets or involve human subjects; all models and benchmarks used are publicly available under

their respective licenses. Our method reduces computation and memory usage, which can lower the energy footprint of large-scale inference. Potential risks stem from the underlying pretrained models (e.g., biases or misuse), which SURGE does not alter. We encourage responsible deployment of VLMs and will release our code to facilitate transparent and reproducible research.

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

# A  APPENDIX

## USE OF LARGE LANGUAGE MODELS

We used large language models (LLMs) solely to assist with grammar checking, wording suggestions, and polishing of the manuscript text. No LLMs were used for idea generation, experimental design, analysis, or results.

## A.1  IMPLEMENTATION DETAILS

We implement SURGE as a lightweight masking module between the vision encoder and the language model, requiring no retraining or modification of backbone weights. For **InternVL-3.5-VL (8B)**, we use the official HuggingFace release and insert SURGE after the vision encoder outputs, before token projection to the LLM. For **Video-LLaVA-Qwen (7B)**, we build on the HuggingFace Transformers implementation of LLaVA-Video, applying SURGE to patch embeddings from the SigLIP-SO400M vision encoder before they are passed into the multimodal backbone. For **Qwen2.5-VL (7B)**, we extend the HuggingFace Transformers classes for the vision transformer and multimodal generation to incorporate SURGE masking in the visual embedding pipeline. For query-aware focusing (SURGE\*), we add a CLIP relevance scorer, applied only to candidate peak frames.

For baseline comparisons, we adopt **FastV** (Chen et al., 2024a) (attention-based pruning) and **AKS** (Tang et al., 2025) (adaptive keyframe selection), using their official open-source implementations within the HuggingFace/Transformers ecosystem. FastV is integrated on **Video-LLaVA-Qwen** via its attention pruning hooks, while AKS performs frame selection upstream of the vision encoder and is applicable to all models.

All benchmarks are evaluated using *VLMEvalKit* (Duan et al., 2024), a standardized open-source toolkit for multimodal evaluation. **We adopt its official protocols, prompt templates, and decoding settings** for Video-MME, MLVU, MMBench-Video, TempCompass and LongVideoBench, ensuring consistent and comparable results. All models, baselines, and toolkits are open-source, and we will release our SURGE/SURGE\* modules and wrapper code for each model upon acceptance. **Results are averaged over 5 random seeds** ($\{41, 79, 138, 534, 963\}$) **for robustness.**

## A.2  BASELINE METHODS

**FastV.** FastV (Chen et al., 2024a) targets inefficiencies in VLM attention by observing that image and video tokens receive vanishingly small attention in deeper layers of models like LLaVA and Qwen-VL. To reduce redundant computation, FastV adaptively prunes visual tokens after a chosen layer, guided by their average attention scores. This plug-and-play pruning avoids both self-attention and FFN costs on discarded tokens, gaining large FLOP savings with little performance drop. Similar attention-guided pruning strategies exist (Zhang et al., 2024; Ye et al., 2025; Huang et al., 2024), but they rely on internal attention maps that may not be exposed in deployed systems. In contrast, SURGE is *training-free and attention-agnostic*: it measures novelty directly via token prediction error, enabling pruning without accessing internal gradients or hidden states. This makes SURGE more robust across backbones and applicable even when attention maps are inaccessible.

**Adaptive Keyframe Sampling (AKS).** AKS (Tang et al., 2025) reduces temporal redundancy by selecting a subset of video frames prior to encoding. It balances *relevance* (frame–query similarity) and *coverage* (temporal diversity) through a recursive judge-and-split strategy, yielding high-quality keyframes under fixed context budgets. While highly effective for long-video QA, AKS operates only at the frame level, meaning all patch tokens from selected frames are preserved. SURGE is complementary: it operates *within frames*, pruning predictable patches after encoding. Combining AKS and SURGE leverages both temporal and token-level sparsity, balancing efficiency and accuracy across compute budgets.

**Random Pruning.** Random pruning removes a fixed fraction of visual tokens uniformly at random. Despite its simplicity, it is often a surprisingly competitive baseline: Wen et al. (Wen et al., 2025) show that random selection or simple pooling can match or even outperform attention-based pruning methods such as FastV and SparseVLM on several benchmarks, due to position bias and instability

in attention-based importance scores. More broadly, Transformers are known to be robust to random token dropping at low-to-moderate ratios, as demonstrated by Kim et al. (Kim et al., 2022), which highlights the redundancy and sparsity inherent in visual tokens. However, because random pruning ignores content, it risks discarding critical information as pruning increases. In contrast, SURGE explicitly measures temporal predictability: it prunes only redundant tokens while retaining novel ones, achieving efficiency without sacrificing essential content.

## A.3 BENCHMARKS AND MODELS: DETAILS AND JUSTIFICATION

### A.3.1 BENCHMARKS

We evaluate on five public video–language benchmarks that together span *breadth* (general QA), *depth* (temporal reasoning), and *length* (long-context). This diversity avoids redundancy and ensures SURGE is tested under complementary stressors.

(1) **Video-MME** (Fu et al., 2025): covers short and long clips with multimodal inputs and diverse QA types, and is widely used as a general evaluation suite for VLMs. We include it both in main results and as the basis for FLOP/latency profiling (Fig. 5), since efficiency patterns are primarily model-driven and do not require redundant measurements across multiple datasets.

(2) **MLVU** (Zhou et al., 2025): a long-video, multi-task benchmark reporting M-Avg/G-Avg across heterogeneous tasks such as needle QA, grounding, and summarization. Because it stresses both extended context and task variety, we use it broadly: in main results, in ablations/hyperparameter sweeps, and for long-context scaling (Fig. 4), where its long videos make it the natural choice to probe robustness under extreme input lengths.

(3) **MMBench-Video (MMB-V)** (Fang et al., 2024): emphasizes multi-step and compositional reasoning on multi-shot videos, complementing MLVU's multi-task design with a focus on logical consistency and stepwise inference. We use it in main results and ablations with **Video-LLaVA-Qwen**, since its free-form QA format is particularly sensitive to token loss.

(4) **TempCompass** (Liu et al., 2024): isolates temporal reasoning (order, speed, duration) by constructing nearly identical clips that differ only in motion attributes. We include it to verify that pruning does not compromise event chronology, a weakness of many sparsity methods. It appears in main results and ablations as a targeted "stress test" for SURGE.

(5) **LongVideoBench (LVB)** (Wu et al., 2024): targets very long videos with referring/retrieval-style questions, stressing extreme long-range reasoning. We use it in main results to demonstrate SURGE's scalability, and in hyperparameter studies with **Qwen2.5-VL**, since only models with extended context can operate in this regime.

**Benchmark coverage.** All five benchmarks are included in the main results with all three models. Ablations and hyperparameters focus on **Video-LLaVA-Qwen** with MLVU, TempCompass, and MMB-V, as these sets are most diagnostic for pruning behavior. Efficiency analysis is done on Video-MME with **Qwen2.5-VL**, and long-context scaling on MLVU with **Qwen2.5-VL**, since these are the only realistic pairings for those analyses.

### A.3.2 MODELS

We select three open-source VLMs that represent distinct integration styles and compute regimes, ensuring that SURGE is evaluated across both high-capacity and practical backbones.

(1) **InternVL-3.5-VL (8B)** (Chen et al., 2024b): a flagship open model with strong general VLM performance and adaptive vision encoding. We include it in the main results to show SURGE remains effective on a high-capacity backbone at the frontier of open performance.

(2) **Video-LLaVA-Qwen (7B)** (Li et al., 2024a): a widely used LLaVA-Video variant coupling a Qwen LLM with a SigLIP encoder, capped at 64 frames. Because of its popularity and stable inference pipeline, we use it extensively for ablations and hyperparameter sweeps, making comparisons with prior sparsity methods more direct.

(3) **Qwen2.5-VL (7B)** (Bai et al., 2025; Wang et al., 2024a): the latest long-context VL model with efficient vision encoding and a $\sim$131k token window. It is the only backbone among the three that

can run extended contexts beyond 230 frames on standard hardware, which is why most efficiency breakdowns, scaling studies, and token–accuracy trade-offs are conducted on it.

**Model coverage.** InternVL-3.5-VL represents a flagship, high-capacity baseline; Video-LLaVA-Qwen covers a video-specialized, widely adopted LLaVA-style architecture; and Qwen2.5-VL represents the state of the art in long-context efficiency. This spread justifies why not all models appear in every experiment, while ensuring SURGE is validated across distinct backbones and deployment scenarios.

## A.4 Additional Visualizations of SURGE Masking

We provide qualitative examples of SURGE across different video domains. Figures 6–8 show raw and smoothed surprise curves alongside a *subset of the processed frames* for clarity. Red boxes indicate peak events, and shaded patches denote tokens retained by SURGE. In our implementation, the first frame is always treated as full-surprise and fully preserved, since no temporal history exists for prediction. Results demonstrate that SURGE highlights novel content across natural, animated, and lecture-style videos.

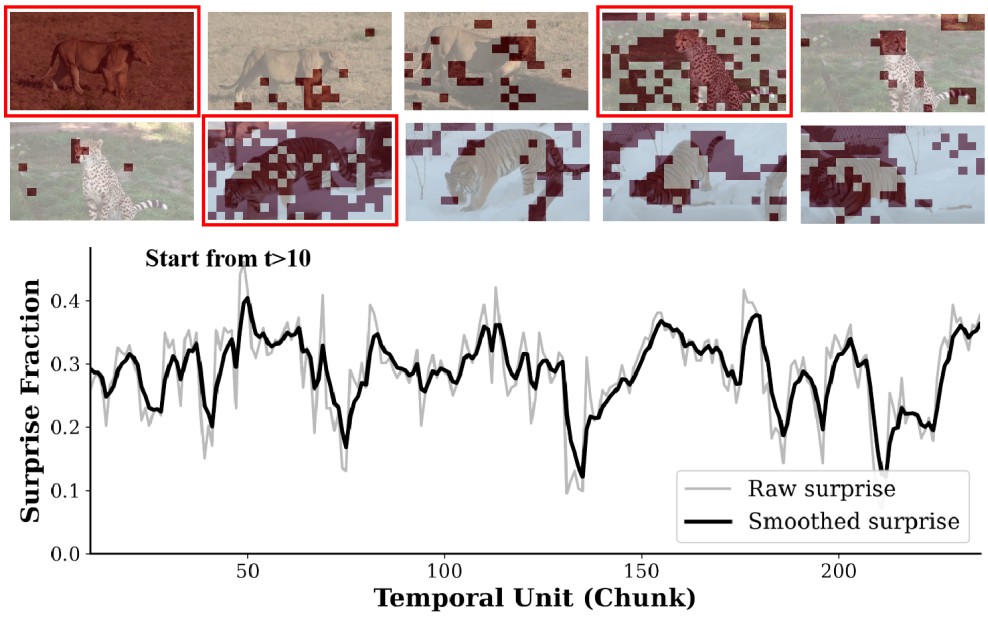

Figure 6: SURGE visualization on an **animal documentary** clip. SURGE emphasizes novel motion events (e.g., cheetah appearance, tiger interaction), with smoothed surprise peaks aligning with key scene changes.

## A.5 Additional Qualitative Examples

We further illustrate SURGE and SURGE⋆ on video–QA tasks (Figs. 9–11). Across these examples, SURGE/SURGE⋆ perform on par with the base model, successfully retaining key evidence tokens. For clarity, we display only a subset of processed frames; in SURGE⋆, red boxes mark peak-event selections, and we show the first frame of each event.

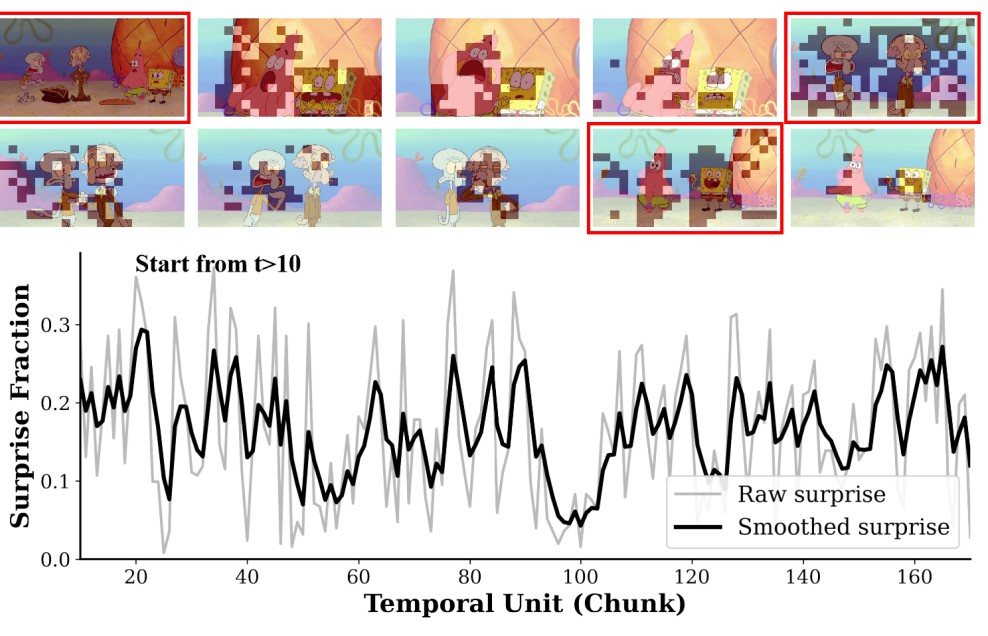

Figure 7: SURGE visualization on a **cartoon** (SpongeBob) clip. Surprise spikes capture exaggerated expressions and character interactions, while redundant frames are pruned.

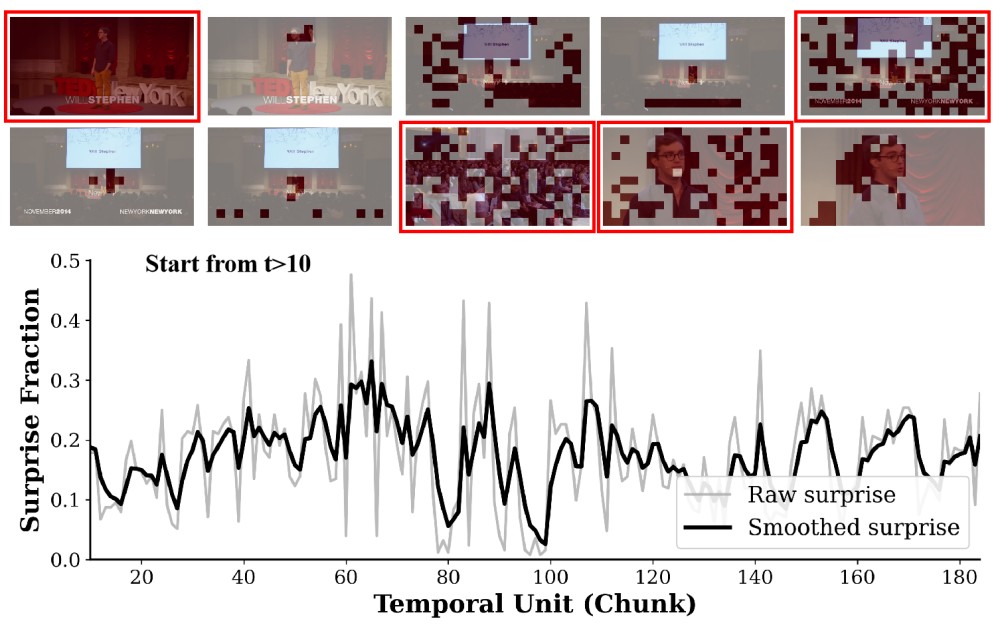

Figure 8: SURGE visualization on a **TED talk** clip. Peaks correspond to speaker appearance changes and audience reactions, whereas static slides yield low surprise.

Q: As depicted in the video, which tool is not necessary to make a rubber band car?

A. Straw    **B. Pencil**    C. Scissors  D. Hammer

**Base Model**: *B. Pencil*    **SURGE**: *B. Pencil*    **SURGE\***: *B. Pencil*

Figure 9: Both SURGE and SURGE* retain the key content and match the base model's correct prediction.

Q: When demonstrating the Germany modern Christmas tree is initially decorated with apples, candles and berries, which kind of the decoration has the largest number?

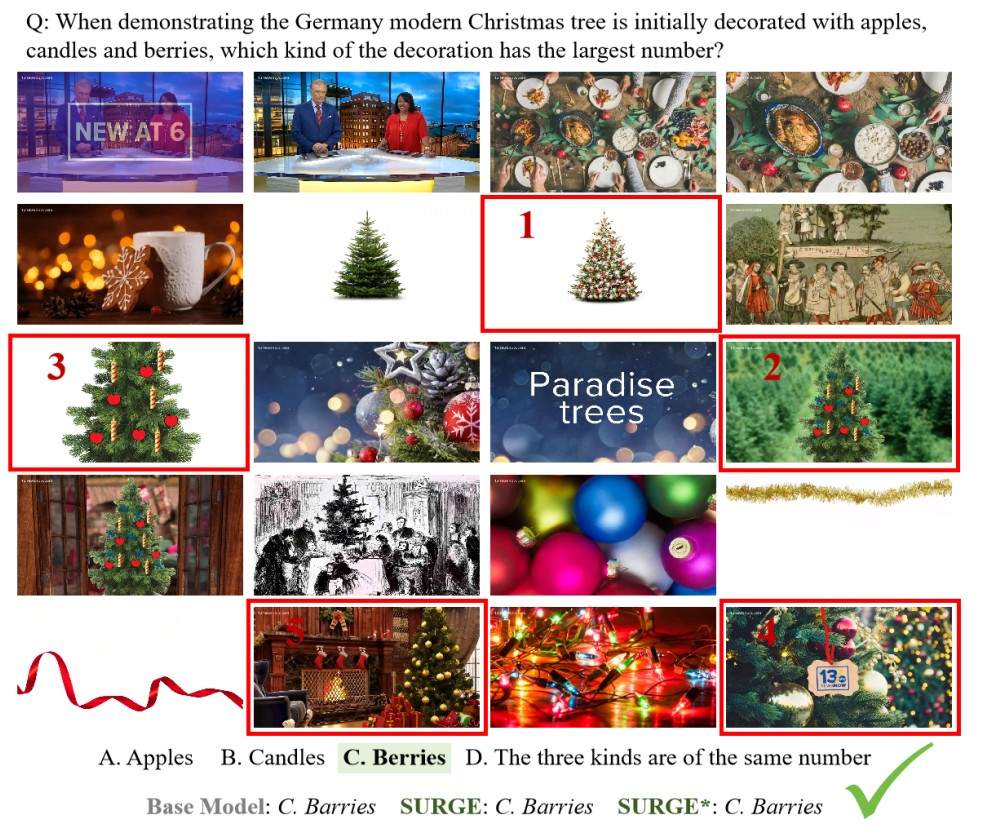

A. Apples    B. Candles    **C. Berries**    D. The three kinds are of the same number

**Base Model**: *C. Barries*    **SURGE**: *C. Barries*    **SURGE\***: *C. Barries*

Figure 10: This task requires counting the dominant decoration type. SURGE and SURGE⋆ preserve key frames showing dense berries, enabling them to match the base model's correct answer despite heavy pruning.

Q: Which of the following reasons motivated the archaeologists to excavate the tomb?

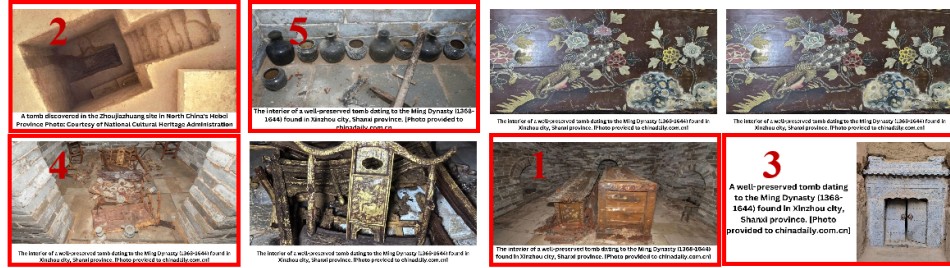

A. Because it's from Ming Dynasty and of specific archaeological significance.
B. Because a new railway line will be built nearby.
C. Because there were treasures inside the tomb.
D. Highway realignment.

**Base Model**: *A*    **SURGE**: *C*    **SURGE\***: *A*    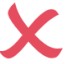

Figure 11: The correct reason ("highway realignment") is non-visual and not inferable from frames alone. All models, including the base, fail here.

