# OpenReview forum: "SURGE: Surprise-Guided Token Reduction for Efficient Video Understanding with VLMs"
_ICLR.cc/2026/Conference — ICLR 2026 Poster_

### Official Review · Reviewer_ky99 · 2025-10-16

**Soundness:** 3
**Presentation:** 2
**Contribution:** 2
**Rating:** 4
**Confidence:** 4

**Summary:**

This paper introduces SURGE, a training-free and model-agnostic method for improving the efficiency of VLMs. The authors propose defining "surprise" as the prediction error of a visual token based on its recent history. A lightweight constant-velocity temporal predictor, enhanced with affine drift correction and variance normalization, is used to calculate a surprise score for each token at each frame.

Based on these scores, SURGE employs a two-stage process to create a sparse spatio-temporal mask, including token pruning and event segmentation. For query-based tasks, the method can be extended to SURGE⋆, which uses CLIP to rank these key events by their relevance to the text query, further focusing computation. The authors conduct extensive experiments on three state-of-the-art VLMs (InternVL-3.5-VL, Video-LLaVA-Qwen, Qwen2.5-VL) across five diverse video understanding benchmarks. The results show that SURGE can reduce token counts by up to 7x and prefill costs by 86–98%, while maintaining accuracy within ±1 percentage point of the full-token baseline, demonstrating its effectiveness and practicality.

**Strengths:**

- Soundness of the core idea: The central contribution is the formulation of "surprise" as a criterion for token retention. Using a simple, physics-inspired constant-velocity predictor in the token embedding space to measure novelty is both original in this context and conceptually elegant. It provides a principled way to distinguish new information from redundant content, moving beyond heuristics like attention weights which can be noisy or biased.

- Practicality: SURGE is training-free and backbone-agnostic. This is a major strength, as it can be plugged into virtually any ViT-based VLM without costly retraining or architectural modifications. This makes the method highly practical and lowers the barrier to adoption for improving inference efficiency.

- Impressive empirical performance: The paper demonstrates substantial efficiency gains while maintaining accuracy.

**Weaknesses:**

- The submission conflicts with the format requirement: "The table number and title always appear before the table." I am not sure whether it should be a reason for a desk rejection.

- The paper claims "negligible overhead" for the SURGE module, but this is not sufficiently quantified.

- The method introduces too many new hyperparameters but does not justify how these hyperparameters are chosen and how they influence the performance.

- The constant-velocity model with affine drift correction is a core component. While effective for many scenarios, it has inherent limitations. It may struggle with complex, non-linear motions such as rapid rotations, non-rigid deformations, or fast zooms, potentially leading to inaccurate surprise scores. Besides, the temporal predictor uses only the displacement from the immediate previous step. This limited temporal window makes the model susceptible to noise and unable to capture more complex, non-linear dynamics such as acceleration or periodic motions.

- There have been many token-level compression methods proposed in the past year, such as ToMe in the related work, and [1] [2]. Please see the survey [3] for more details. The paper should compare with these methods.

- The proposed method operates on the patch embeddings after they have been processed by the vision encoder. While it effectively reduces the computational load on the subsequent large language model, it does not address the significant upstream cost of encoding every single frame of a long video. For extremely long videos, the vision encoder itself can become a major bottleneck. It would be valuable to explore whether the core idea of "surprise" could be adapted to operate at a lower level—perhaps on raw pixels or early-layer features—to enable dynamic frame dropping or resolution adjustment before the expensive full ViT forward pass.

- The method relies on a "global percentile" threshold and peak-based event segmentation, both of which seem to require processing the entire video clip offline to establish a global context of surprise scores. This "two-pass" nature makes its direct application to true streaming or real-time video analysis challenging, where decisions must be made causally and on-the-fly with only past and present information.

[1] ViLAMP: Scaling Video-Language Models to 10K Frames via Hierarchical Differential Distillation
[2] Divprune: Diversitybased visual token pruning for large multimodal models
[3] When Tokens Talk Too Much: A Survey of Multimodal Long-Context Token Compression across Images, Videos, and Audios

**Questions:**

Please see weakness.

---

> ### Author Response · Authors · 2025-11-18
>
> We thank the reviewer for their positive comments on our core idea, the practicality of our design and the empirical results. Below we respond to each concern:
>
> **Q1 Formatting of tables.** Thank you for pointing out the formatting issue. We are sorry about this typesetting mistake. In the updated manuscript, we have corrected the table formatting so that every table follows the required convention, and we have also checked the rest of the paper for similar layout issues.
>
> **Q2 Quantifying SURGE's overhead and relation to other compression methods.**
> Thank you for raising this concern. In the original submission we described SURGE's cost as "negligible" based mainly on FLOPs and latency curves for a single model. Following the suggestions from Reviewers xZY7 and XSco, we have now extended both the comparisons and the efficiency analysis using the open-source implementations of recent training-free methods (DyCoKe, LLaVA-Scissor, VisionZip, FastV) in the **lmms-eval** framework. To make the results directly comparable, we follow the same setup as these works: 32-frame inputs and a LLaVA-OneVision backbone, and we also report results on **Qwen2.5-VL-7B** under the same 32-frame protocol. The corresponding accuracy tables are shown in Q5's response. Here we summarize the efficiency side. For **latency overhead** on 32-frame clips, we obtain:
> | Method | Latency Overhead | Breakdown                                                                            |
> | :----------------- | ---------------: | :----------------------------------------------------------------------------------- |
> | LLaVA-Scissor      |         433.9 ms | spatial compression 254.6 ms + temporal compression 165.5 ms + token merging 13.7 ms |
> | FastV              |          96.3 ms | –                                                                                    |
> | DyCoKe             |         88.73 ms | –                                                                                    |
> | VisionZip          |           5.8 ms | dominant token 0.16 ms + token merging 5.65 ms                                       |
> | Surprise+VisionZip |           7.5 ms | surprise scoring 1.24 ms + token merging 6.23 ms                                     |
> | SURGE          |       **4.7 ms** | surprise scoring 1.31 ms + token pruning 3.43 ms                                     |
> | SURGE* (K=5) |     507.4 ms | frame index and reload 453.61 ms + CLIP scoring 53.74 ms                               |
>
> These results show that SURGE adds only a few milliseconds of overhead per 32-frame clip. Its cost is smaller than methods that do heavy processing inside the vision or language tower (such as LLaVA-Scissor, DyCoKe). Note, in our Qwen2-based FastV adaptation, we have to replace the k-1 attention layer with an eager implementation to read out attention scores, which increases latency. Ideally, with native support this overhead could be reduced, but such implementations are not yet available. On the other hand, SURGE* adds substantial extra cost due to frame indexing/reload and CLIP scoring, but is intended for offline or long-context use where FLOPs and memory savings matter more than per-query latency, while SURGE serves as the default low-overhead variant.
>
> For **FLOPs** on VideoMME with 32 frames, using the same estimation method as DyCoKe and LLaVA-Scissor (following LLaVA-Scissor's Eq. 9), we obtain approximately:
> | Method (25% retention)    | FLOPs (T) |
> | :-------------------------------- | --------: |
> | Baseline (full tokens)            | **43.31** |
> | LLaVA-Scissor (τ=0.9, ε=0.05) |     12.37 |
> | FastV (k=3)                     |     16.13 |
> | DyCoKe (l=3, k=0.5)           |     14.41 |
> | VisionZip                         |     12.83 |
> | SURGE                         | **12.77** |
> | SURGE* (K=5)              |  **7.22** |
>
> Because SURGE and SURGE* use percentile-based thresholds, the actual retention can be slightly above or below 25% (e.g., 24–26%), which explains the small FLOPs differences between SURGE and LLaVA-Scissor (pruninig before the language model). Even so, SURGE matches or slightly improves on the FLOPs of other token-reduction methods at comparable retention while keeping latency very low, and SURGE* with CLIP pass roughly halves the FLOPs again relative to SURGE at the price of higher latency. We also note, as discussed in our response to Reviewer xZY7, that when re-implementing the FLOPs estimator from DyCoKe and LLaVA-Scissor we found a GFLOPs-to-TFLOPs conversion error in our earlier code. The values in Fig. 5(a) of the original submission should be divided by an additional factor of 10, and this has been corrected in the updated version.

---

> ### Author Response · Authors · 2025-11-18
>
> **Q3 Hyperparameters (γ, Δ, K, ρ).** We understand the concern about the number of hyperparameters and agree that their influence should be made more explicit. In the submitted version, we already ablated two of the key knobs in Sec. 5.1:
> * **ρ** (retention ratio) in Table 2, showing that SURGE maintains accuracy within about ±1 point of the baseline over a wide range of token budgets, and degrades much less than random pruning.
> * **K** (number of events kept in SURGE*) in Table 3, showing that performance quickly stabilizes once K reaches a moderate value (around 5), and we then fix K=5 for all SURGE* experiments.
>
> The other two hyperparameters, **γ** (EMA smoothing) and **Δ** (minimum separation between surprise peaks), were introduced to give finer control over temporal smoothing and event segmentation. In the initial experiments we selected their defaults using a broad random search over γ in [0.7, 0.95] and Δ in [4, 12]. We now include their sweeps to directly address the reviewer's question.
>
> For **SURGE** at ρ=25% on Qwen2.5-VL-7B:
>
> | γ                      | VideoMME | LongVideoBench | TempCompass |
> | :--------------------- | -------: | -------------: | ----------: |
> | Baseline (full tokens) | **62.2** |       **60.0** |    **70.5** |
> | 0.70                   |     60.1 |           59.1 |        70.1 |
> | 0.75                   |     60.1 |           59.3 |        70.1 |
> | 0.80                   |     60.4 |       **59.4** |        70.4 |
> | 0.85                   |     60.3 |       **59.4** |    **70.5** |
> | 0.90                   | **60.9** |       **59.4** |    **70.5** |
> | 0.95                   | **60.9** |           59.3 |    **70.5** |
>
> For **SURGE⋆** at ρ=25%, γ=0.9, K=5:
>
> | Δ                      | VideoMME | LongVideoBench | TempCompass |
> | :--------------------- | -------: | -------------: | ----------: |
> | Baseline (full tokens) | **62.2** |       **60.0** |    **70.5** |
> | 4                      |     59.7 |           55.8 |        67.5 |
> | 6                      | **62.1** |           56.1 |    **67.7** |
> | 8                      |     62.0 |       **56.3** |    **67.7** |
> | 10                     |     61.8 |       **56.3** |    **67.7** |
> | 12                     |     60.2 |           56.0 |        67.5 |
>
> These tables show that both **γ** and **Δ** have a broad stable region (e.g., γ in [0.8, 0.95], Δ in [6, 10]): performance varies only slightly within these ranges, and our chosen defaults (γ=0.9, Δ=8) are representative rather than carefully tuned. Combined with the existing ablations on **ρ** and **K**, this indicates that although SURGE has several hyperparameters, in practice we use the same configuration across all datasets and backbones and still obtain consistent performance. We also note that introducing such hyperparameters for finer-grained control (e.g., thresholds, ratios, or layer choices) is common in recent training-free compression methods like DyCoKe, LLaVA-Scissor and VisionZip. We will add these sweep tables to the appendix and clarify this behaviour in the revised manuscript. For more discussion of hyperparameter robustness, we also refer the reviewer to our answer to Reviewer xZY7, Q2.

---

> ### Author Response · Authors · 2025-11-18
>
> **Q4 Constant-velocity model, non-linear motion and short temporal window.** We appreciate the reviewer's concern about the limits of a constant-velocity predictor. Our choice here is not purely heuristic but follows the analysis in Sec. 3. There we assume that (i) natural videos evolve smoothly over time and (ii) the per-patch encoder is differentiable. Under these assumptions, a first-order Taylor expansion of the token trajectory yields the constant-velocity relation in Sec. 3, where the second temporal difference is approximately zero. This directly motivates the causal predictor we use: extrapolating the next token from its most recent displacement in token space.
>
> Importantly, SURGE **does not** aim to estimate physical motion (e.g., exact optical flow). The predictor is only used to measure deviation from expected evolution (surprise), not to recover the underlying dynamics. Any significant departure from constant-velocity behaviour (like acceleration, deformation, occlusion or fast camera changes) simply increases the residual and thus the surprise score. In practice, this means complex motion tends to make tokens more likely to be retained, rather than pruned. We use only the immediate previous step to keep the predictor causal, training-free, and lightweight. To reduce the failure modes the reviewer mentions, SURGE already incorporates:
> * **Affine drift correction**, which fits and subtracts a global affine trend in token space, so that coherent camera motion does not dominate surprise.
> * **Variance normalization**, which rescales residuals by a running variance, so tokens that always move strongly do not receive high surprise unless their behaviour actually changes.
>
> Fig. 1 provides a sanity check for this design: after drift removal and variance normalization, the normalized surprise curves align well with true content changes and correlate with pixel-level dissimilarity, even under global motion. Empirically, if the constant-velocity, short-window model struggled badly with complex dynamics, we would expect clear accuracy drops on challenging long-video datasets. Instead, in our experiments (including new experimental results in Q5 tables), SURGE and SURGE* remain within a narrow range of the full-token baseline while discarding a large fraction of tokens, and are competitive with or better than other training-free pruning methods at comparable budgets. This indicates that the predictor is robust for SURGE’s purpose of identifying informative tokens. We will clarify these design choices and their limitations in the revised manuscript, and we agree that exploring higher-order or learned temporal predictors, while keeping the method lightweight, is an interesting direction for future work. We kindly refer the reviewer to our relevant responses to Reviewer xZY7 Q1.

---

> > ### Comment · Reviewer_ky99 · 2025-11-24
> >
> > So why not use  pixel dissimilarity to guide vision token compression?

---

> > > ### Author Response · Authors · 2025-11-24
> > >
> > > Thank you for raising this and it is a very natural question. Our choice to compute surprise in token space is mainly about staying aligned with what the VLM actually uses and keeping the signal stable.
> > >
> > > Firstly, the model never reasons over pixels directly. The language model part only ever sees these vision tokens, not pixel values. Measuring surprise directly on these tokens lets us detect "what changed" in the representation that the model really uses to answer questions, rather than in the raw image space.
> > >
> > > Also, ViT features already mix space and time in a nontrivial way. After a few layers, each token aggregates information from a spatial region and neighboring frames. At that point, a naive pixel-wise difference between frames is a noisy proxy for how much the internal state has changed: small pixel shifts can be irrelevant, and modest pixel changes can flip semantics. This is also why most existing compression/pruning methods in our experiments (e.g., attention-based pruning, token merging) operate in feature/token space instead of raw pixels.
> > >
> > > Third, in practice, token-space surprise behaves better while still reflecting pixel changes. In Fig. 1, we use pixel dissimilarity as a sanity check: after drift removal and variance normalization, our normalized surprise curves line up well with true content changes and correlate with pixel differences. So we keep the "actual novelty" of pixel change while discarding irrelevant factors. Doing all this in token space also keeps SURGE training-free and lightweight, without adding an extra pixel-level model on top of the existing encoder.
> > >
> > > We appreciate the reviewer bringing up this point and would be very happy to further discuss or clarify any aspect of this choice.

---

> > > ### Author Response · Authors · 2025-11-27
> > >
> > > We would like to kindly follow up on our rebuttal and again thank the reviewer for the thoughtful suggestions and constructive discussion. Please let us know if any further clarification would be helpful.

---

> > > > ### Comment · Reviewer_ky99 · 2025-11-27
> > > >
> > > > Overall, this is a solid study. I appreciate the concept of "surprise," which may be a potential alternative for "salient/important information". These are the reasons why I do not give a lower rating. The authors try to give a theoretical derivation of "surprise", which I think is sound. But I am still confused about the link between "surprise" defined in the paper and traditional "salient information" for VLMs. That is, is the information selected by the surprise score salient? Nowadays, many token compression techniques have been proposed for video-language models. And there are many heuristics for defining saliency. I am worried whether this paper is just yet another heuristic.

---

> > > > > ### Author Response · Authors · 2025-11-27
> > > > >
> > > > > Thank you very much for the thoughtful follow-up and for keeping an open mind about the "surprise" idea. We fully understand the concern: with so many token compression methods and saliency heuristics around, it's important to be clear whether this is just "one more heuristic".
> > > > >
> > > > > In our view, SURGE treats saliency as "what is new given the past context", rather than “what looks important in isolation”. In long videos, many tokens are on-topic but redundant (persistent backgrounds, repeated views, etc.). Our surprise score is defined as deviation from the expected token trajectory given recent history. A token is kept because it changes relative to what has already been seen, not because its feature is large at a single step. This makes surprise a temporal, context-aware notion of importance, which is conceptually different from, for example, using one-step attention magnitude as a saliency score.
> > > > >
> > > > > Furthermore, we fully agree that any practical signal is still a proxy for "truly salient for the final answer", including ours. The distinction we aim for is that many existing methods define saliency via attention weights, CLIP similarity or learned selectors, often relying on internal attention maps, extra heads or trained modules plus tuned thresholds. In contrast, our surprise score is derived from local dynamics of the encoder features (Sec. 3). So SURGE’s signal is tightly tied to a concrete, causal prediction model of how tokens should evolve if nothing new is happening, rather than being an opaque hand-crafted scalar.
> > > > >
> > > > > Empirically, we see that this behaves like a useful saliency signal for VLMs without needing extra training. In our visualizations, high-surprise regions concentrate on object motions, interactions, and scene transitions, while static backgrounds are heavily pruned. Quantitatively, our comprehensive experiments already show encouraging performance. The Surprise+VisionZip experiment further supports this: replacing VisionZip’s attention-based dominant-token rule with surprise yields comparable or slightly improved performance, suggesting that surprise is not only a viable importance signal on its own but also easy to reuse inside existing token-merging pipelines. This **flexibility** is important in practice because it means SURGE’s scoring mechanism can benefit other compression architectures without retraining or architectural changes.
> > > > >
> > > > > Now, we will make this positioning explicit: SURGE does not claim to perfectly capture "semantic saliency" in all cases, but proposes a prediction-error–based, dynamics-grounded proxy for informativeness that is different in nature from existing heuristics and that empirically behaves as a stable and effective selection signal for long-video VLMs. We really appreciate your pushing us to articulate this link more clearly.

---

> ### Author Response · Authors · 2025-11-18
>
> **Q5 Comparison to recent token-level compression methods.** Thank you for raising this. Combining the suggestions from all reviewers, we conducted a more comprehensive study of recent token-level compression methods and, in the rebuttal, added comparisons with the open-sourced, training-free video token methods that are most directly comparable to SURGE: **DyCoKe**, **VisionZip** and **LLaVA-Scissor**. These works represent the current SOTA for training-free pruning/merging on long video inputs, and their official implementations are built on the **lmms-eval** framework with **32-frame** inputs and the **LLaVA-OneVision**. For fair and directly comparable evaluation, we adopt exactly this setup and also run the same 32-frame protocol on **LLaVA-OneVision** and **Qwen2.5-VL-7B**.
>
> #### **(A) Qwen2.5-VL-7B**
> | Method (Hyperparameters) | EgoSchema | NextQA | VideoMME | LongVideoBench |
> | :---------------------------------- | --------: | -----: | --------: | --------------: |
> | **Baseline**                        | **58.4**  | **74.6** | **61.4** | **58.8** |
> | **25% Retention** | | | | |
> | FastV (k=3)                    | 56.9      | 72.5   | 58.8     | 56.3          |
> | VisionZip                    | 57.9      | 74.4   | 60.7     | 58.3          |
> | Surprise+VisionZip           | 58.1      | 74.5   | 59.1     | **58.4**         |
> | SURGE                        | 58.2      | **74.7**   | 60.9     | 58.3          |
> | SURGE* (~12.5%, K=5)               | **58.3**      | 74.6   | **61.3**     | 58.2          |
> | **50% Retention** | | | | |
> | FastV (k=3)                    | 57.4      | 74.3   | 60.2     | 58.1          |
> | VisionZip                    | **58.7**  | 74.9   | 61.1     | 58.6          |
> | Surprise+VisionZip          | 58.6      | 74.9   | 59.7     | **59.0**      |
> | SURGE                       | **58.7**  | **75.1** | 61.3   | 58.6          |
> | SURGE* (~26%, K=5)                 | 58.7      | 75.0   | **61.4** | 58.4          |
>
>
> #### **(B) LLaVA-OneVision**
>
> | Method (Hyperparameters) | EgoSchema | NextQA | VideoMME | LongVideoBench |
> | :---------------------------------- | --------: | -----: | --------: | --------------: |
> | **Baseline**                        | **60.3**  | **79.4** | **58.5** | **56.6**       |
> | **10% Retention** | | | | |
> | DyCoKe (l=3, k=0.5)           | **60.3**      | 78.8   | 57.9     | **56.3**          |
> | LLaVA-Scissor (τ=0.86, ε=0.05) | 59.4     | 78.5   | 57.0     | 56.0          |
> | FastV (k=3)                   | 57.3      | 73.9   | 52.3     | 54.7          |
> | SURGE                       | 59.6      | **79.1**   | 57.8     | 55.9          |
> | SURGE* (~5%, K=5)                  | 60.1      | 79.0   | **58.0**     | 56.2          |
> | **25% Retention** | | | | |
> | DyCoKe (l=3, k=0.5)           | **60.4**  | 78.8   | 57.9     | **56.5**          |
> | LLaVA-Scissor (τ=0.9, ε=0.05) | 59.7     | 79.1   | 57.1     | 56.3          |
> | FastV (k=3)                   | 58.8      | 77.9   | 56.6     | 55.8          |
> | SURGE                       | 60.2      | **79.3**   | **58.7** | **56.5**         |
> | SURGE* (~12.5%, K=5)               | 60.2      | 79.0   | 58.2     | **56.5**          |
> | **50% Retention** | | | | |
> | DyCoKe (l=3, k=0.5)           | **60.4**  | 78.8   | 58.0     | 56.5          |
> | LLaVA-Scissor (τ=0.925, ε=0.05) | 60.1    | **79.4** | 58.1   | **56.8**      |
> | FastV (k=3)                   | 59.9      | 78.7   | 57.7     | 56.3          |
> | SURGE                         | 60.3      | 79.3   | **58.7** | **56.8**      |
> | SURGE* (~26%, K=5)                 | 60.2      | **79.4** | 58.4   | 56.5          |
>
> The tables above show that, under this shared setting, **SURGE** and **SURGE*** achieve accuracy that is comparable to or slightly better than these methods at similar retention ratios, while remaining training-free, backbone-agnostic and lightweight to integrate. We also evaluate a **"Surprise+VisionZip"** variant, where SURGE's surprise scores replace VisionZip's attention-based dominant-token selection, and observe comparable or improved performance at the same budgets, indicating that our surprise signal is effective even when plugged into existing token-merging pipelines. In the revised manuscript, we will also expand the related-work section to explicitly discuss **ToMe**, **ViLAMP**, **DivPrune**, and the survey suggested by the reviewer. For a more detailed analysis of these new experimental results, we kindly refer the reviewer to our response to Reviewer ZWDa.

---

> ### Author Response · Authors · 2025-11-18
>
> **Q6 Operating only after the vision encoder and not reducing encoder cost.** We agree with the reviewer that, for extremely long videos, the vision encoder itself can become a major bottleneck, and that adapting the "surprise" idea earlier in the pipeline (e.g., for frame dropping or resolution control) is a valuable direction. In this work, we place SURGE pruning **after** the vision encoder and **before** the LLM for two reasons:
> 1. Modern video VLMs use quite different vision backbones and temporal modules (e.g., temporal patch aggregation, varying tokenization schemes, or multi-stage fusion). In our preliminary attempts to apply surprise directly on raw patches or early-layer features, we found that doing so in a non-invasive, training-free way quickly became model-specific: a strategy that works for one backbone (e.g., with a particular temporal merging design) does not transfer cleanly to another without modifying internal blocks. This conflicts with our goal of a plug-and-play, backbone-agnostic method.
>
> 2. We also experimented with pruning at very early stages (e.g., selecting "surprising" patches before the full ViT stack). Even with careful heuristics, this produced unstable and unrelevant responses. Our interpretation is that early layers carry low-level information and have less redundancy: aggressive removal there changes the context of attention markedly and propagates downstream, making the final token embeddings highly sensitive to small early perturbations. In contrast, at the vision-encoder output, temporal redundancy is much higher and the high-level tokens are more robust, so pruning there (and before the LLM) is much safer. This is also where the KV-cache cost becomes a major practical bottleneck, which is what we target in this paper.
>
> We fully agree that pushing surprise-based reasoning deeper into the vision stack (e.g., on early-layer features for frame dropping or dynamic resolution) is an important next step, and we will mention this explicitly as a promising future direction building on SURGE. Our current design focuses on the common, architecture-agnostic interface (vision encoder outputs) to keep the method training-free, widely applicable, and stable across different VLMs. Again, we appreciate the reviewer's insightful comment.
>
> **Q7 Streaming, global percentile, and "two-pass" usage.** We thank the reviewer for raising this point. Our current description can indeed give the impression that SURGE requires seeing the entire clip before making decisions. In fact, the core SURGE masking is defined in a causal way and does not require a second pass over the video. As stated in Sec. 4.1, we define the buffer as "the full clip offline, or the observed prefix in streaming", i.e., surprise scores are computed per frame using only past information, and the global percentile can be estimated over the scores seen so far (e.g., via a running or approximate quantile). The binary mask is then applied directly after the vision encoder and before the LLM in the same forward pass, without re-encoding frames.
>
> The reviewer's concern is more applicable to SURGE*. In SURGE*, we first construct a surprise curve over time, detect peaks, and then run CLIP on the corresponding key frames to rank events and select the top-K windows. This query-aware focusing is indeed better suited to offline or batched scenarios (e.g., long video retrieval). Even there, however, we only re-index the selected frames for CLIP, the vision encoder is not run twice.
>
> In the revision, we will clarify this distinction: (i) SURGE itself can be used in a single-pass, causal manner by computing surprise and percentile thresholds over the observed prefix, which makes it compatible with streaming or low-latency settings; (ii) SURGE* is intended as an optional, query-aware extension for offline or long-context use. We agree that explicitly exploring online approximations to global percentiles and peak detection (e.g., sliding-window surprise curves) is an interesting future direction, and we will highlight this as a natural extension of SURGE.

---

### Official Review · Reviewer_XSco · 2025-10-29

**Soundness:** 3
**Presentation:** 3
**Contribution:** 2
**Rating:** 6
**Confidence:** 4

**Summary:**

This paper introduces SURGE, a training-free, and model-agnostic method designed to enhance the long video understanding by dynamically reducing redundant input tokens. It uses a simple feature predictor to estimate the next frame's visual features based on the current one. If the surprise score is low, the corresponding tokens are pruned, significantly reducing the VLM's computational load. Then it refines with CLIP-based query relevance. SURGE reduces tokens by up to 7× and prefill cost by 86–98%, while keeping accuracy within ±1 point of full-token baselines.

**Strengths:**

+ The method is training-free and thus can be applied to any LVLMs.
+ It reduces tokens by up to 7× and prefill cost by 86–98% while preserve the accuracy within ±1 point.
+ The authors provide comprehensive results based on several LVLMs across multiple video understanding benchmarks.

**Weaknesses:**

+ The paper lacks a discussion of related work on video token compression and frame selection methods [1–5].
+ Although the proposed method improves efficiency, it does not yield a clear accuracy gain. Ideally, removing redundant tokens should free up contextual capacity to accommodate more informative content, thereby enhancing the model’s ability to answer queries.
+ The proposed surprise scoring mechanism is intended to capture pixel-level changes, yet it relies on CLIP features trained in semantic space.
+ Missing comparison with training-free token pruning methods [1,2,5].

[1] DyCoke: Dynamic Compression of Tokens for Fast Video Large Language Models, CVPR 2025.
[2] VisionZip: Longer is Better but Not Necessary in Vision Language Models, CVPR 2025.
[3] Chat-UniVi: Unified Visual Representation Empowers Large Language Models with Image and Video Understanding, CVPR 2024.
[4] LongVU: Spatiotemporal Adaptive Compression for Long Video-Language Understanding, ICML 2025.
[5] BOLT: Boost Large Vision-Language Model Without Training for Long-form Video Understanding, CVPR 2025.

**Questions:**

+ Previous studies have shown that sampling more frames generally leads to better performance. Ideally, if redundant tokens are removed while key frames are preserved, the model should achieve improved results under the same context budget. Have the authors tested their method under the same context budget as the base model to verify whether token redundancy reduction effectively enhances context utilization and leads to better performance?
+ As shown in Figure 2, why is the surprise score computed on the visual features after the linear projection? Since the projection aligns the features with the LLM’s input space rather than the pure visual space, wouldn’t this affect the intended focus? Have the authors conducted any experiments using features extracted directly from the ViT backbone?
+ Additionally, is the frame–text similarity based on CLIP features also calculated after the linear projection?
+ How does the proposed method perform compared with DyCoKe, VisionZip, and BOLT?
+ The proposed surprise scoring mechanism is primarily designed to capture pixel-level changes. However, CLIP features are trained using a vision–language contrastive objective in a semantic space, which may not align well with this goal. Have the authors considered using vision-centric models, such as DINO, to compute the surprise scores instead?

---

> ### Author Response · Authors · 2025-11-18
>
> We thank the reviewer for highlighting SURGE's adaptability to different VLMs and the breadth of our experimental evaluation across multiple models and benchmarks. Below we respond to each of the raised concerns:
>
> **Q1 More related work discussion.** We appreciate the reviewer for pointing this out. In the revised manuscript, we have explicitly added DyCoKe, VisionZip, Chat-UniVi, LongVU and BOLT to the related work section (highlighted in red).
>
> **Q2 Accuracy gains, redundancy, and context budget.** We thank the reviewer for bringing up this interesting point, and we agree that ideally, removing redundant tokens should free up context for more informative content and can sometimes lead to accuracy gains. In our results, we do observe such improvements in several cases: SURGE* slightly exceeds the full-token baseline on long-video benchmarks such as MLVU and LongVideoBench, and in our new experiments (see the tables in Q4 and our responses to Reviewer ZWDa and Reviewer xZY7 Q5) SURGE/SURGE* match or modestly improve over the baseline on datasets like NextQA, EgoSchema, VideoMME and LongVideoBench at 25–50% retention. However, both prior work and our extended comparisons indicate that training-free compression methods typically deliver stable performance rather than large accuracy gains over strong full-token VLMs, which is consistent with what we observe.
>
> For clarification, the main goal of SURGE is to provide a good token–accuracy trade-off and better effective context usage, rather than to guarantee a large accuracy boost at a fixed short context. Tables 1–2 in the paper and the new lmms-eval results show that SURGE and SURGE* keep accuracy performance while discarding a large fraction of tokens. Under a fixed memory or context budget, our long-context experiments (Fig. 4) show that SURGE and especially SURGE* allow the model to process many more frames before hitting context limits, with competitive or slightly improved performance compared to dense or truncated baselines. This combination of stable accuracy under strong pruning and extended usable context is the primary benefit of SURGE.

---

> ### Author Response · Authors · 2025-11-18
>
> **Q3 Surprise scoring, CLIP space and vision-centric features.** We thank the reviewer for raising this point and would like to clarify two misunderstandings:
> 1. **Where the surprise score is computed.**
>     Surprise is not computed in CLIP space and not on features after the linear projection into the LLM. As described in Sec. 3 and made explicit in Appendix A.1, we insert SURGE *"after the vision encoder outputs, before token projection to the LLM."* In practice, the surprise score is computed directly on the vision-tower patch embeddings (ViT features) produced by the VLM's backbone, and the resulting mask is then applied to the projected tokens right before they are fed into the LLM. We realize that Fig. 2 may be misleading here: the SURGE block is drawn after the linear projector, which can give the impression that surprise is computed post-projection. In the revision, we have clarifed in the text that only the masking happens at that stage, while the surprise scores themselves are always computed on the raw vision-encoder tokens.
>
>     Furthermore, CLIP is used only in SURGE* as an optional, query-aware event selector: we use CLIP to score the similarity between the text query and a small set of candidate key frames, in order to choose which temporal windows to keep. This CLIP step never enters the definition of token surprise itself.
> 2. **"Pixel-level change" vs. feature-space surprise.**
>    When we describe SURGE as capturing "pixel-level changes", our intent is to detect changes in the underlying visual content via the encoder features, not to operate on raw pixels. Sec. 3 motivates this by showing that, under smooth video dynamics and a differentiable vision encoder, token trajectories are predictable and deviations from this behaviour naturally indicate content changes. Fig. 1 then provides empirical evidence: after drift removal and variance normalization, the normalized surprise curves closely track pixel dissimilarity for example patches, with peaks aligned to actual changes and a roughly monotonic relationship. This shows that the surprise statistic defined in feature space is strongly correlated with true visual change.
>
> Our new experiments further support the quality of this signal. In particular, when we plug SURGE's surprise scores into VisionZip and keep its merging stage unchanged ("Surprise+VisionZip"), this variant achieves comparable or slightly better accuracy than the original VisionZip at the same token budgets (see the Qwen2.5-VL table in Q4). This indicates that the surprise scores derived from the VLM's own vision encoder form a strong and useful importance signal, even when integrated into an existing token-merging pipeline.
>
> Regarding the suggestion to use vision-centric models such as DINO, we agree this is an interesting direction. In this work, we deliberately reuse the VLM's own vision backbone to keep SURGE training-free, model-agnostic, and lightweight, without introducing an additional encoder. As future work, we will consider to explore using self-supervised vision models like DINO to analyse the surprise-selected frames and potentially refine temporal prioritization.

---

> ### Author Response · Authors · 2025-11-18
>
> **Q4 Comparison with DyCoKe, VisionZip and BOLT.** Thank you for pointing this out. In the new experiments, we now explicitly include **VisionZip**, **DyCoKe**, and **LLaVA-Scissor**, together with other recent training-free baselines such as **FastV**. Following their released implementations, we adopt the same lmms-eval setup with 32-frame inputs and the LLaVA-OneVision backbone (and also report results on Qwen2.5-VL-7B under the same protocol). As shown in the tables below, under this shared configuration SURGE and SURGE* achieve accuracy that is comparable to or better than these methods at similar retention ratios, while remaining training-free, model-agnostic and lightweight to integrate.
>
> As mentioned, we also evaluate a **"Surprise+VisionZip"** variant, where SURGE's surprise scores replace VisionZip's attention-based dominant-token selection. This variant attains comparable or slightly improved accuracy at the same budgets (see the Qwen2.5-VL-7B table), indicating that our surprise signal is also effective when plugged into existing token-merging pipelines. For *BOLT: Boost Large Vision-Language Model Without Training for Long-form Video Understanding* (CVPR 2025), we didn't include its results because the code is not publicly available. In our view, as a training-free, query-guided frame selection method, BOLT plays a similar role to AKS and is naturally complementary to SURGE: BOLT (or AKS) can first select informative frames, and SURGE can then prune redundant tokens within those frames. For a more detailed discussion of these expanded comparisons and the associated FLOPs/latency analysis, we refer the reviewer to our response to **Reviewer XSco Q3/4**.
>
> #### **(A) Qwen2.5-VL-7B**
> | Method (Hyperparameters) | EgoSchema | NextQA | VideoMME | LongVideoBench |
> | :---------------------------------- | --------: | -----: | --------: | --------------: |
> | **Baseline**                        | **58.4**  | **74.6** | **61.4** | **58.8** |
> | **25% Retention** | | | | |
> | FastV (k=3)                    | 56.9      | 72.5   | 58.8     | 56.3          |
> | VisionZip                    | 57.9      | 74.4   | 60.7     | 58.3          |
> | Surprise+VisionZip           | 58.1      | 74.5   | 59.1     | **58.4**         |
> | SURGE                        | 58.2      | **74.7**   | 60.9     | 58.3          |
> | SURGE* (~12.5%, K=5)               | **58.3**      | 74.6   | **61.3**     | 58.2          |
> | **50% Retention** | | | | |
> | FastV (k=3)                    | 57.4      | 74.3   | 60.2     | 58.1          |
> | VisionZip                    | **58.7**  | 74.9   | 61.1     | 58.6          |
> | Surprise+VisionZip          | 58.6      | 74.9   | 59.7     | **59.0**      |
> | SURGE                       | **58.7**  | **75.1** | 61.3   | 58.6          |
> | SURGE* (~26%, K=5)                 | 58.7      | 75.0   | **61.4** | 58.4          |
>
> #### **(B) LLaVA-OneVision**
> | Method (Hyperparameters) | EgoSchema | NextQA | VideoMME | LongVideoBench |
> | :---------------------------------- | --------: | -----: | --------: | --------------: |
> | **Baseline**                        | **60.3**  | **79.4** | **58.5** | **56.6**       |
> | **10% Retention** | | | | |
> | DyCoKe (l=3, k=0.5)           | **60.3**      | 78.8   | 57.9     | **56.3**          |
> | LLaVA-Scissor (τ=0.86, ε=0.05) | 59.4     | 78.5   | 57.0     | 56.0          |
> | FastV (k=3)                   | 57.3      | 73.9   | 52.3     | 54.7          |
> | SURGE                       | 59.6      | **79.1**   | 57.8     | 55.9          |
> | SURGE* (~5%, K=5)                  | 60.1      | 79.0   | **58.0**     | 56.2          |
> | **25% Retention** | | | | |
> | DyCoKe (l=3, k=0.5)           | **60.4**  | 78.8   | 57.9     | **56.5**          |
> | LLaVA-Scissor (τ=0.9, ε=0.05) | 59.7     | 79.1   | 57.1     | 56.3          |
> | FastV (k=3)                   | 58.8      | 77.9   | 56.6     | 55.8          |
> | SURGE                       | 60.2      | **79.3**   | **58.7** | **56.5**         |
> | SURGE* (~12.5%, K=5)               | 60.2      | 79.0   | 58.2     | **56.5**          |
> | **50% Retention** | | | | |
> | DyCoKe (l=3, k=0.5)           | **60.4**  | 78.8   | 58.0     | 56.5          |
> | LLaVA-Scissor (τ=0.925, ε=0.05) | 60.1    | **79.4** | 58.1   | **56.8**      |
> | FastV (k=3)                   | 59.9      | 78.7   | 57.7     | 56.3          |
> | SURGE                         | 60.3      | 79.3   | **58.7** | **56.8**      |
> | SURGE* (~26%, K=5)                 | 60.2      | **79.4** | 58.4   | 56.5          |

---

> ### Comment · Reviewer_XSco · 2025-11-25
>
> Thanks for the detailed response. The rebuttal adds useful comparisons with other training-free methods and clarifies where and in which space the surprise score is computed. However, several concerns remain unresolved.
>
> **Comparison under the same maximum frames setting**
>
> The response to Q4 does not report the base model’s official results. If we keep the same context budget used in the original evaluations. For example, Qwen2.5-VL reports 65.1 on VideoMME and 65.0 on EgoSchema with roughly a 20k context window with 256 frames as input, can the proposed method outperform these official scores? In principle, using the same context size but selecting more informative frames (rather than uniform sampling) should allow the method to surpass the base model’s reported performance.
>
> **Lower-than-expected base model performance**
>
> In Table 1, Qwen2.5-VL uses 41,590 tokens to represent 64 frames, and the authors report 65.8 on MLVU and 62.2 on VideoMME. In practice, reducing the max-pixel setting allows the model to achieve around 65.1 on VideoMME and 70.2 on MLVU within a 20k-token budget for 256 frames input. This shows the author didn't carefully adjust the token per frame to reflect the base model’s actual performance.
>
> Therefore, to ensure a valid comparison, the base model’s actual performance should be reflected, i.e., using 256 frames within a 20k context window, which matches the model’s official results. After establishing this baseline, you can then reduce to 64 frames and apply compression to enable a fair comparison with training-free methods. Otherwise, I am afraid that the performance gain in table 1 benefits from the surprise-based selecting or benefits from just adjusting the token-per-frame setting.

---

> > ### Author Response · Authors · 2025-11-26
> >
> > ## Update on 256-frame setting.
> > Following the reviewer's suggestion, we have now run Qwen2.5-VL-7B on VideoMME with **256 frames** and a ~23k-token context window in **VLMEvalKit**. Under this setting, the full-token baseline reaches **66.6**, which is in line with (and even slightly higher than) the 65.1 score mentioned by the reviewer. This confirms that the evaluation tool and configuration we use are fully capable of matching the base model’s actual performance and do not depress the baseline. On top of this stronger baseline, **SURGE (ρ = 50%)** achieves **67.1** and **SURGE (ρ = 25%)** reaches **66.2**, showing that SURGE can match or even slightly improve over the accurately configured 256-frame baseline. This further supports that our comparisons are valid and that the gains we report come from SURGE itself rather than from a weakened base configuration. We are continuously running the SURGE* under this as well, but those experiments require more time to complete.
> >
> > We hope that these results help resolve the reviewer's concern, and we sincerely appreciate the reviewer's careful feedback and constructive suggestions, which have led to a more thorough evaluation.
> >
> > Additionally, these results can be reproduced in VLMEvalKit with:
> >
> > ```bash
> > torchrun --nproc-per-node=8 run.py \
> >   --data Video-MME_256frame \
> >   --model Qwen2.5-VL-7B-Instruct \
> >   --verbose
> > ```
> >
> > **Note.** The `Video-MME_256frame` setting is obtained by a minimal edit to `video_dataset_config.py` in VLMEvalKit (increasing VideoMME's sampled frames to 256); all other parts of the VLMEvalKit pipeline remain unchanged.

---

> > > ### Comment · Reviewer_XSco · 2025-11-26
> > >
> > > Thank the authors for updating the results in the 256-frame setting. This resolves my remaining concern. Theoretically, selecting the most informative frames while reducing redundancy can enhance the base model’s performance. The reported gain from 66.6 to 67.1 by applying SURGE, with half of the tokens reducted. This provides evidence supporting the effectiveness of the proposed method.
> > >
> > > It would be valuable to including this result in the main. I would like to keep my positive assessment for this work.

---

> > > > ### Author Response · Authors · 2025-11-26
> > > >
> > > > We sincerely thank the reviewer for the thoughtful follow-up and are very glad that the updated experiments fully address the remaining concern. We appreciate the reviewer's positive assessment and encouraging comments on the effectiveness of SURGE. As suggested, we will include the new 256-frame results in the main paper to make the presentation clearer and further strengthen our contribution.
> > > >
> > > > We are genuinely grateful for the reviewer's constructive feedback throughout the process, which has helped us improve and refine the work.

---

> ### Author Response · Authors · 2025-11-26
>
> We greatly appreciate the reviewer's thoughtful follow-up and would like to respectfully clarify a few misunderstandings around the baseline setup.
>
> First, all of our original results use a single, fixed protocol based on **VLMEvalKit**, which is now a very common tool for evaluating open LVLMs. Under this setting, Qwen2.5-VL with 64 frames and the default resolution gives **62.2** on VideoMME in our runs. On the public VLMEvalKit leaderboard, Qwen2.5-VL 7B reports about **62.8** on the same benchmark, which is very close and shows that our setup is consistent and reproducible rather than "under-tuned". We did not deliberately lower the base model's performance. Instead, we strictly adhered to a widely adopted standardized evaluation process rather than attempting to replicate every single "official" configuration option.
>
> The configuration the reviewer refers to is a **different** evaluation setting: more frames, different max-pixel/resolution and a different context limit. In the current ecosystem there are many such pipelines (official scripts, VLMEvalKit, lmms-eval, internal code, etc.), and they do not always match each other exactly. Reproducing and re-tuning Qwen2.5-VL under all of these configurations is beyond the scope of this work. Instead, we follow the standard practice in recent efficiency papers: (1) pick one or two standardized toolchains (VLMEvalKit in the paper and lmms-eval in the rebuttal), (2) keep the protocol fixed, and (3) only change the token-reduction method.
>
> Within each framework, the comparison is strictly fair. In **VLMEvalKit**, the full-token baseline and all SURGE / SURGE* runs use exactly the same number of frames, resolution and pre/post-processing. Table 2 further shows that random pruning at the same retention ratios hurts performance much more than SURGE, which rules out the explanation that our gains come purely from changing the token budget or the tokens-per-frame setting.
>
> In the rebuttal, the used **lmms-eval** has been widely adopted by recent training-free methods, such as **DyCoKe** (CVPR 2025), **VisionZip** (CVPR 2025) and **BOLT** (CVPR 2025). These works are all implemented and benchmarked there under its unified setup. We adopt that exact setting and run all methods (including SURGE and SURGE*) under the same configuration. In this framework, our baseline results match those reported in the **BOLT** paper and are very close to the numbers reported in **DyCoKe** and **VisionZip**, which further validates the evaluation setup.
>
> *To address the core concern*, we respectfully disagree that not matching a specific "official" 256-frame, 20k-token configuration might make our comparison invalid. In current LVLM practice, what matters for a fair efficiency comparison is that all methods share the *same* evaluation pipeline, not that this pipeline coincides with every configuration reported in the base model paper. In our case, the VLMEvalKit baselines are correctly configured and closely match the public VLMEvalKit leaderboard, and the lmms-eval baselines match the numbers reported in recent SOTA work. On top of these well-established baselines, SURGE is applied without changing the underlying protocol and consistently achieves strong token–accuracy trade-offs. We see this as a valid and standard way to evaluate an efficiency method, and we will make this rationale explicit in the revised version so that the role of SURGE is clearly separated from the choice of evaluation pipeline.
>
> We are currently running 256-frame VideoMME baselines using both **VLMEvalKit** and **lmms-eval**, and will include these results here once they are available. Given the strong agreement we already observe with public leaderboards and prior work, we expect the numbers to be close to the accuracy mentioned by the reviewer. We again sincerely thank the reviewer for the careful reading and thoughtful feedback, which have helped us sharpen both our experiments and the presentation of this work, and we look forward to continued discussion.

---

### Official Review · Reviewer_xZY7 · 2025-10-31

**Soundness:** 3
**Presentation:** 3
**Contribution:** 3
**Rating:** 4
**Confidence:** 4

**Summary:**

language models named SURGE. Specifically, SURGE includes two core modules: Surprise Estimation and Spatio-temporal Masking. Surprise Estimation leverages a constant-velocity predictor in token space with global affine drift removal and variance normalization to measure temporal predictability, where high-surprise tokens are identified by large prediction errors indicating novel content. Spatio-temporal Masking applies global percentile thresholding to retain the most surprising tokens and aggregates scores over time to produce a surprise curve for key event segmentation.

The experimental evaluation in this paper assessed the performance of the proposed SURGE on five public benchmarks (Video-MME, MLVU, MMBench-Video, TempCompass, and LongVideoBench), comparing it with various approaches.

**Strengths:**

- The definition of token surprise is considered novel.
- The method’s performance has been validated across multiple base models, enhancing the credibility of the approach.

**Weaknesses:**

- SURGE models temporal dynamics using constant-velocity prediction plus affine drift fitting. Compared to optical flow or more advanced temporal prediction methods, this simple predictor may be insufficient for complex motions or non-linear changes (e.g., rapid acceleration/deceleration, severe local deformations). Could there be missed detections of key tokens in high-complexity videos?
- SURGE introduces several hyperparameters (γ, ∆, K, ρ), but the paper provides limited ablation discussion on their effects. Although Sec. 5.3 claims stability within reasonable ranges; in different tasks or video distributions, these parameters may require retuning.
- SURGE⋆ adds an additional CLIP scoring step. Does this introduce extra computational overhead?
- The FLOPs and latency analysis lack comparison with other methods.
- The paper lacks comparison against other approaches that reduce tokens before the LLM stage, such as VisionZip [1] or LLaVA-Scissor [2].

[1] VisionZip: Longer is Better but Not Necessary in Vision Language Models

[2] LLaVA-Scissor: Token Compression with Semantic Connected Components for Video LLMs

**Questions:**

- In SURGE, it seems the decision to retain a token is based solely on its individual surprise score. If part of an object’s tokens are removed due to low surprise, could the remaining tokens lose semantic completeness and break the spatial continuity of the object representation? Why is there only a retain-or-delete operation at the token level, without a merging/fusion mechanism to preserve semantic integrity?

---

> ### Author Response · Authors · 2025-11-18
>
> We thank the reviewer for recognizing the novelty of our token-surprise formulation and for noting that our experiments across multiple base models support the credibility of the approach. Below we respond to each of the raised concerns:
>
> **Q1 Temporal dynamics and constant-velocity prediction.** We appreciate the reviewer's concern about temporal modeling. Our choice of a constant-velocity predictor in token space is grounded in the analysis in Sec. 3, rather than being a heuristic. Under standard assumptions for video and ViT-based encoders, we model frames as evolving smoothly over time and the per-patch encoder (f_j) as differentiable. A first-order Taylor expansion of the token trajectory then yields Eq. (1) and Eq. (2), where the second temporal difference is approximately zero. This corresponds to a constant-velocity prior in token space and naturally motivates the causal predictor in Eq. (3), which extrapolates the next token from its recent displacement. The predictor is therefore aligned with the smooth dynamics of encoder features.
>
> More importantly, SURGE does not attempt to recover detailed physical motion (e.g., optical flow). The predictor is used only to measure how much a token deviates from its expected evolution, i.e., to quantify surprise. If a token's representation evolves in a predictable way, the residual stays small and the surprise score remains low; if its trajectory changes sharply due to new content, acceleration, deformation or occlusion, the residual becomes large and the token is marked as highly surprising. Any departure from constant-velocity behavior increases the prediction error and thus the surprise score. In this sense, complex motion tends to make tokens more likely to be retained, not more likely to be missed. The subsequent drift removal and variance normalization further refine this signal: the global affine drift fitting (Eq. 4) removes coherent camera motion and other scene-wide trends, and the variance normalization (Eq. 5) calibrates each token's residual by its own historical variability. Fig. 1 illustrates that the resulting normalized surprise aligns well with actual content changes and correlates approximately linearly with pixel dissimilarity, indicating that the measure tracks true visual change in a stable way.
>
> Also, our experiments support this design. In the ablations (Sec. 5.3), removing the temporal predictor and using simple frame differences leads to the largest degradation among the tested variants, confirming that the constant-velocity predictor contributes meaningfully to performance. In addition, our extended evaluation now includes **NextQA** and **EgoSchema**, which both contain complex, egocentric and temporally challenging videos. On these benchmarks, SURGE and SURGE* maintain accuracy very close to the full-token baseline across a range of token budgets, and remain competitive with or better than other training-free pruning methods. This suggests that, in practice, the proposed predictor does not miss key events even in high-complexity videos, and that the theoretically motivated surprise signal behaves robustly across diverse datasets.

---

> ### Author Response · Authors · 2025-11-18
>
> **Q2 Hyperparameters (γ, Δ, K, ρ).** We agree that these hyperparameters should be discussed more explicitly, and that in principle the optimal setting may vary across tasks or video distributions. However, in practice, we find that SURGE and SURGE* behave stably over a broad range of values, so we use a single configuration across all datasets and models. As in many recent training-free methods (e.g., DyCoKe, LLaVA-Scissor), having several hyperparameters is common, but in our case we do not rely on careful per-dataset tuning.
>
> These hyperparameters fall into two groups: (ρ) is the token retention ratio (standard for token-reduction methods), while (γ, Δ, K) control the temporal smoothing and segment selection in SURGE / SURGE*. In the original submission, we performed a broad random search for (γ in [0.7, 0.95]) and (Δ in [4, 12]) on representative datasets, found a wide stable region, and then fixed a single configuration for all experiments. We did not include the detailed sweeps in the submission, which we now add them below for clarity. For SURGE at (ρ=25%) (Qwen2.5-VL-7B):
>
> | γ               | VideoMME | LongVideoBench | TempCompass |
> | :--------------------- | -------: | -------------: | ----------: |
> | Baseline (full tokens) | **62.2** |       **60.0** |    **70.5** |
> | 0.70                   |     60.1 |           59.1 |        70.1 |
> | 0.75                   |     60.1 |           59.3 |        70.1 |
> | 0.80                   |     60.4 |           **59.4** |        70.4 |
> | 0.85                   |     60.3 |           **59.4** |        **70.5** |
> | 0.90                   | **60.9** |           **59.4** |        **70.5** |
> | 0.95                   | **60.9** |           59.3 |        **70.5** |
>
> For SURGE* at (ρ=25%), (γ=0.9, K=5):
>
> | Δ               |  VideoMME | LongVideoBench | TempCompass |
> | :--------------------- | --------: | -------------: | ----------: |
> | Baseline (full tokens) |  **62.2** |       **60.0** |    **70.5** |
> | 4                      |      59.7 |           55.8 |        67.5 |
> | 6                      | **62.1** |           56.1 |        **67.7** |
> | 8                      | 62.0 |           **56.3** |        **67.7** |
> | 10                     |      61.8 |           **56.3** |        **67.7** |
> | 12                     |      60.2 |           56.0 |        67.5 |
>
> These scans show that both (γ) and (Δ) have a broad stable region (e.g., γ in [0.8, 0.95], Δ in [6,10]), and our chosen setting (γ=0.9, Δ=8) is representative rather than finely tuned. For (K), Table 3 in the paper already shows that performance is stable once (K>=5). Across all benchmarks, including the new results on NextQA, EgoSchema, and additional backbones, we reuse this single hyperparameter configuration and still obtain consistent performance. We will add these sweep tables to the appendix and clarify this behavior in the revision.

---

> ### Author Response · Authors · 2025-11-18
>
> **Q3/4 CLIP overhead and FLOPs/latency comparison.** Yes, the additional CLIP scoring in SURGE* introduces extra computational overhead. This variant is intended for scenarios where aggressive FLOPs and memory reduction on very long videos is more important than per-query latency (e.g., offline retrieval or long-context scoring). In contrast, the main SURGE variant is designed to be the default low-overhead option. We have now added FLOPs and latency measurements for SURGE, SURGE* and recent training-free baselines (see Q4 or our response to Reviewer ZWDa), under the same lmms-eval and 32-frame setting as in our extended accuracy experiments.
>
> For latency overhead, we obtain:
> | Method | Latency Overhead | Breakdown                                                                            |
> | :----------------- | ---------------: | :----------------------------------------------------------------------------------- |
> | LLaVA-Scissor      |         433.9 ms | spatial compression 254.6 ms + temporal compression 165.5 ms + token merging 13.7 ms |
> | FastV              |          96.3 ms | –                                                                                    |
> | DyCoKe             |         88.73 ms | –                                                                                    |
> | VisionZip          |           5.8 ms | dominant token 0.16 ms + token merging 5.65 ms                                       |
> | Surprise+VisionZip |           7.5 ms | surprise scoring 1.24 ms + token merging 6.23 ms                                     |
> | SURGE          |       **4.7 ms** | surprise scoring 1.31 ms + token pruning 3.43 ms                                     |
> | SURGE* (K=5) |     507.4 ms | frame index and reload 453.61 ms + CLIP scoring 53.74 ms                               |
>
> These numbers show that SURGE adds only a few milliseconds of overhead and is in the same regime as VisionZip, while being much lighter than methods that heavily operate inside the vision or language tower. For example, LLaVA-Scissor does reduce tokens before the LLM but incurs substantial cost in the vision tower. FastV and DyCoKe also has higher overhead because they depend on access to internal LM attention. Note, in our Qwen2 adaptation of FastV, we replace the k–1 attention layer with an eager version to access attention scores, which further increases latency. With an ideal native implementation its cost could be lower, but such support is not yet available.
>
> For FLOPs on VideoMME, we now use the same estimation method as DyCoKe and LLaVA-Scissor (following LLaVA-Scissor's Eq. 9), we obtain:
>
> | Method (25% retention)    | FLOPs (T) |
> | :-------------------------------- | --------: |
> | Baseline (full tokens)            | **43.31** |
> | LLaVA-Scissor (τ=0.9, ε=0.05) |     12.37 |
> | FastV (k=3)                     |     16.13 |
> | DyCoKe (l=3, k=0.5)           |     14.41 |
> | VisionZip                         |     12.83 |
> | SURGE                         | **12.77** |
> | SURGE* (K=5)              |  **7.22** |
>
> Because SURGE and SURGE* rely on percentile-based selection, the actual retention ratio can deviate slightly from an exact 25% (e.g., 24–26%), which explains the small FLOPs differences between methods at the same budget. Even with this, the overhead remains minimal: SURGE matches or slightly improves on the FLOPs of other token-reduction methods at comparable retention while keeping latency very low. SURGE* roughly halves the FLOPs again compared to SURGE/VisionZip, at the cost of higher per-query latency due to reload frames and CLIP scoring.
>
> We will clarify in the revision that SURGE is intended as the default low-overhead variant for interactive use, while SURGE* targets memory- and FLOPs-constrained long-context scenarios where this trade-off is desirable. We also thank the reviewer for prompting this analysis. In re-implementing the FLOPs estimator used in DyCoKe and LLaVA-Scissor, we found a GFLOPs-to-TFLOPs conversion error in our earlier code; the values in Fig. 5a should be divided by an additional factor of 10, and we have corrected this in the revision.

---

> ### Author Response · Authors · 2025-11-18
>
> **Q5 Comparison to VisionZip and LLaVA-Scissor.** Thank you for pointing this out. In the new experiments, we now explicitly include both VisionZip and LLaVA-Scissor as baselines, along with additional recent training-free methods suggested by other reviewers, such as DyCoKe and FastV. Following their released implementations, we adopt the same lmms-eval setup with 32-frame inputs and the LLaVA-OneVision backbone. As shown in the expanded tables above, under this shared protocol SURGE and SURGE* achieve accuracy that is comparable to or better than these methods at similar retention ratios, while remaining training-free, model-agnostic and lightweight to integrate. We also evaluate a "Surprise+VisionZip" variant, where SURGE's surprise scores replace VisionZip's attention-based dominant-token selection, and observe comparable or slightly improved accuracy at the same budgets, indicating that our surprise signal is also effective when plugged into existing token-merging pipelines. For a more detailed discussion of these expanded comparisons, we kindly refer the reviewer to our response to Reviewer ZWDa. Together with the FLOPs and latency measurements provided above, these results indicate that SURGE offers a robust and efficient alternative among training-free token-reduction methods.
> #### **(A) Qwen2.5-VL-7B**
> | Method (Hyperparameters) | EgoSchema | NextQA | VideoMME | LongVideoBench |
> | :---------------------------------- | --------: | -----: | --------: | --------------: |
> | **Baseline**                        | **58.4**  | **74.6** | **61.4** | **58.8** |
> | **25% Retention** | | | | |
> | FastV (k=3)                    | 56.9      | 72.5   | 58.8     | 56.3          |
> | VisionZip                    | 57.9      | 74.4   | 60.7     | 58.3          |
> | Surprise+VisionZip           | 58.1      | 74.5   | 59.1     | **58.4**         |
> | SURGE                        | 58.2      | **74.7**   | 60.9     | 58.3          |
> | SURGE* (~12.5%, K=5)               | **58.3**      | 74.6   | **61.3**     | 58.2          |
> | **50% Retention** | | | | |
> | FastV (k=3)                    | 57.4      | 74.3   | 60.2     | 58.1          |
> | VisionZip                    | **58.7**  | 74.9   | 61.1     | 58.6          |
> | Surprise+VisionZip          | 58.6      | 74.9   | 59.7     | **59.0**      |
> | SURGE                       | **58.7**  | **75.1** | 61.3   | 58.6          |
> | SURGE* (~26%, K=5)                 | 58.7      | 75.0   | **61.4** | 58.4          |
>
> #### **(B) LLaVA-OneVision**
>
> | Method (Hyperparameters) | EgoSchema | NextQA | VideoMME | LongVideoBench |
> | :---------------------------------- | --------: | -----: | --------: | --------------: |
> | **Baseline**                        | **60.3**  | **79.4** | **58.5** | **56.6**       |
> | **10% Retention** | | | | |
> | DyCoKe (l=3, k=0.5)           | **60.3**      | 78.8   | 57.9     | **56.3**          |
> | LLaVA-Scissor (τ=0.86, ε=0.05) | 59.4     | 78.5   | 57.0     | 56.0          |
> | FastV (k=3)                   | 57.3      | 73.9   | 52.3     | 54.7          |
> | SURGE                       | 59.6      | **79.1**   | 57.8     | 55.9          |
> | SURGE* (~5%, K=5)                  | 60.1      | 79.0   | **58.0**     | 56.2          |
> | **25% Retention** | | | | |
> | DyCoKe (l=3, k=0.5)           | **60.4**  | 78.8   | 57.9     | **56.5**          |
> | LLaVA-Scissor (τ=0.9, ε=0.05) | 59.7     | 79.1   | 57.1     | 56.3          |
> | FastV (k=3)                   | 58.8      | 77.9   | 56.6     | 55.8          |
> | SURGE                       | 60.2      | **79.3**   | **58.7** | **56.5**         |
> | SURGE* (~12.5%, K=5)               | 60.2      | 79.0   | 58.2     | **56.5**          |
> | **50% Retention** | | | | |
> | DyCoKe (l=3, k=0.5)           | **60.4**  | 78.8   | 58.0     | 56.5          |
> | LLaVA-Scissor (τ=0.925, ε=0.05) | 60.1    | **79.4** | 58.1   | **56.8**      |
> | FastV (k=3)                   | 59.9      | 78.7   | 57.7     | 56.3          |
> | SURGE                         | 60.3      | 79.3   | **58.7** | **56.8**      |
> | SURGE* (~26%, K=5)                 | 60.2      | **79.4** | 58.4   | 56.5          |

---

> > ### Comment · Reviewer_xZY7 · 2025-11-26
> >
> > I've read the authors' detailed response and truly appreciate the additional experiments on FLOPs and latency, especially the voluntary inclusion of comparisons with pure vision-token compression methods like LLaVA-Scissor and VisionZip. Most of my concerns have now been addressed, so I am raising my score to 6.

---

> > > ### Author Response · Authors · 2025-11-26
> > >
> > > We sincerely thank the reviewer for the positive reassessment and are glad that the additional comparisons have addressed the main concerns. We appreciate the updated score and the constructive feedback throughout the process.

---

> ### Author Response · Authors · 2025-11-18
>
> **Q6 Token-level pruning, semantic completeness and lack of merging.** We appreciate this concern. In SURGE, the retain/delete decision is indeed taken at the token level, but the surprise scores are not independent per-token signals. They are computed from temporal trajectories in token space with drift removal and variance normalization (Sec. 3), which makes surprise spatially and temporally coherent rather than noisy. As shown in Fig. 1, after detrending and normalization, the surprise curves align with genuine content changes rather than global motion, and are stable across neighbouring patches. In practice, regions of an object that undergo meaningful change tend to share elevated surprise, while static or redundant areas (often background) remain low-surprise, so SURGE keeps a sparse but semantically focused set of tokens rather than arbitrarily fragmenting objects.
>
> Empirically, if SURGE frequently destroyed semantic completeness, we would expect large accuracy drops under strong pruning. However, Table 1 and Table 2 show that on Qwen2.5-VL, SURGE keeps accuracy within about 1% of the full-token baseline across a wide range of retention ratios (down to (ρ=0.10) and even 0.01), whereas random pruning at the same (ρ) causes much larger and unstable degradation (e.g., up to 23.9% at (ρ=0.10)). In our extended experiments, the same behavior holds across multiple backbones (Video-LLaVA-Qwen, Qwen2.5-VL-7B, LLaVA-OneVision) and datasets (including NextQA and EgoSchema), all with a single shared configuration. This cross-model consistency provides further evidence that SURGE's token-level pruning preserves the semantics needed for downstream reasoning.
>
> We chose a hard retain/delete mask rather than an explicit merging/fusion mechanism to keep SURGE training-free, backbone-agnostic and easy to integrate as a drop-in module between the vision encoder and the LLM, without modifying internal blocks or retraining summarizers. This design choice is supported by the fact that the same SURGE module works effectively across diverse VLM architectures in our experiments. That also said, SURGE is fully compatible with merging-based designs. In our new experiments, "Surprise+VisionZip" variant achieves comparable or slightly better accuracy than the original VisionZip at the same budgets (see the Qwen2.5-VL table), which shows that the surprise signal can also serve as an effective importance criterion inside a merging pipeline. Finally, as described in Eq. (8), SURGE* also supports an optional context floor that preserves a small number of low-surprise tokens per unit if one wishes to enforce additional spatial continuity; in the current paper we did not use this floor in the main experiments because the token-level surprise already preserved accuracy well across benchmarks.

---

### Official Review · Reviewer_ZWDa · 2025-11-01

**Soundness:** 2
**Presentation:** 3
**Contribution:** 2
**Rating:** 4
**Confidence:** 4

**Summary:**

In this paper, a training-free and model-agnostic token reduction algorithm, called SURGE, is proposed. The proposed algorithm estimates the surprising scores based on the prediction error of each token from its recent history. Then, it remains the tokens with high surprising score, while prunes the tokens with low surprise. Additionally, it utilizes the CLIP similarities for query-focused applications. Experimental results on video understanding datasets show that the proposed algorithm achieves comparable or even better performance despite a lower computational budget.

**Strengths:**

- This paper is well-written and easy to follow.
- The paper includes sufficient implementation details, which enhances its reproducibility.
- The proposed method is motivated by a solid rationale and is technically well justified.

**Weaknesses:**

- From Table 1, the comparison appears to be limited to a single baseline—fastV + Video-LLaVA-Qwen at one fixed budget. If comparisons with additional methods are difficult to include, it would still be helpful to report results across a broader range of budgets to better illustrate robustness and trade-offs.
- L354-357 mentions that AKS and the proposed SURGE are complementary to each other. However, in Table 1, when comparing the results at similar token budgets, some results slightly improve while others decrease, and the others remain almost unchanged, so the overall performance appears to be roughly comparable.
- It would be beneficial if the experimental results on benchmark datasets, such as Next‑QA or EgoSchema, are included.
- This paper appears to lack a comparison with recent training-free token reduction methods. Including such baselines would help clarify the relative advantages of the proposed approach. For example, there are the following recent works:

    [1] Multi-Granular Spatio-Temporal Token Merging for Training-Free Acceleration of Video LLMs, ICCV 2025

    [2] Video, How Do Your Tokens Merge?, CVPR Workshop 2025

    [3] Don’t Look Twice: Faster Video Transformers with Run-Length Tokenization, NeurIPS 2024

- Typo
  - L49 ”surprise” : the quotation marks appear to be incorrectly oriented.

**Questions:**

My main concern is that the paper lacks sufficient performance comparison with existing token reduction methods. If this issue is properly addressed in the rebuttal, I would be willing to raise my score. Please also refer to the Weaknesses section for my other concerns.

---

> ### Author Response · Authors · 2025-11-18
>
> ### Main Concern: Comparative Breadth and Robustness
>
> We appreciate the reviewer for the positive assessment of our motivation, technical soundness and clarity of presentation. In response to the main concern, we have substantially expanded the comparisons. For the new experiments, we adopted the *lmms-eval* framework because several SOTA training-free baselines, including **DyCoKe**, **LLaVA-Scissor** and **VisionZip** (which have been shown to outperform earlier token-merging/compression approaches such as ToMe), are implemented and benchmarked there. To ensure a fair and directly comparable evaluation, we followed the same setup as these works, including the standard 32-frame input configuration. Most of these methods are built on LLaVA-OneVision, which also provides stronger accuracy than the Video-LLaVA-Qwen model used in our submission. Therefore, we conducted the new comparisons on LLaVA-OneVision for consistency with the existing baselines.
>
> Regarding the methods suggested by the reviewer, we mainly examined the ICCV 2025 STTM implementation, but its code depends on older *transformers* versions that are incompatible with the new FlashAttention 2 and Qwen2 stack, so a quick comparison was not feasible. Finally, to confirm consistency with our original results, we re-evaluated SURGE on VideoMME and LongVideoBench within lmms-eval, and the overall trends match those observed with VLMEvalKit.
>
> #### **(A) Qwen2.5-VL-7B**
> | Method (Hyperparameters) | EgoSchema | NextQA | VideoMME | LongVideoBench |
> | :---------------------------------- | --------: | -----: | --------: | --------------: |
> | **Baseline**                        | **58.4**  | **74.6** | **61.4** | **58.8** |
> | **25% Retention** | | | | |
> | FastV (k=3)                    | 56.9      | 72.5   | 58.8     | 56.3          |
> | VisionZip                    | 57.9      | 74.4   | 60.7     | 58.3          |
> | Surprise+VisionZip           | 58.1      | 74.5   | 59.1     | **58.4**         |
> | SURGE                        | 58.2      | **74.7**   | 60.9     | 58.3          |
> | SURGE* (~12.5%, K=5)               | **58.3**      | 74.6   | **61.3**     | 58.2          |
> | **50% Retention** | | | | |
> | FastV (k=3)                    | 57.4      | 74.3   | 60.2     | 58.1          |
> | VisionZip                    | **58.7**  | 74.9   | 61.1     | 58.6          |
> | Surprise+VisionZip          | 58.6      | 74.9   | 59.7     | **59.0**      |
> | SURGE                       | **58.7**  | **75.1** | 61.3   | 58.6          |
> | SURGE* (~26%, K=5)                 | 58.7      | 75.0   | **61.4** | 58.4          |
>
>
> #### **(B) LLaVA-OneVision**
>
> | Method (Hyperparameters) | EgoSchema | NextQA | VideoMME | LongVideoBench |
> | :---------------------------------- | --------: | -----: | --------: | --------------: |
> | **Baseline**                        | **60.3**  | **79.4** | **58.5** | **56.6**       |
> | **10% Retention** | | | | |
> | DyCoKe (l=3, k=0.5)           | **60.3**      | 78.8   | 57.9     | **56.3**          |
> | LLaVA-Scissor (τ=0.86, ε=0.05) | 59.4     | 78.5   | 57.0     | 56.0          |
> | FastV (k=3)                   | 57.3      | 73.9   | 52.3     | 54.7          |
> | SURGE                       | 59.6      | **79.1**   | 57.8     | 55.9          |
> | SURGE* (~5%, K=5)                  | 60.1      | 79.0   | **58.0**     | 56.2          |
> | **25% Retention** | | | | |
> | DyCoKe (l=3, k=0.5)           | **60.4**  | 78.8   | 57.9     | **56.5**          |
> | LLaVA-Scissor (τ=0.9, ε=0.05) | 59.7     | 79.1   | 57.1     | 56.3          |
> | FastV (k=3)                   | 58.8      | 77.9   | 56.6     | 55.8          |
> | SURGE                       | 60.2      | **79.3**   | **58.7** | **56.5**         |
> | SURGE* (~12.5%, K=5)               | 60.2      | 79.0   | 58.2     | **56.5**          |
> | **50% Retention** | | | | |
> | DyCoKe (l=3, k=0.5)           | **60.4**  | 78.8   | 58.0     | 56.5          |
> | LLaVA-Scissor (τ=0.925, ε=0.05) | 60.1    | **79.4** | 58.1   | **56.8**      |
> | FastV (k=3)                   | 59.9      | 78.7   | 57.7     | 56.3          |
> | SURGE                         | 60.3      | 79.3   | **58.7** | **56.8**      |
> | SURGE* (~26%, K=5)                 | 60.2      | **79.4** | 58.4   | 56.5          |

---

> ### Author Response · Authors · 2025-11-18
>
> **Q1 Limited FastV setting.** In the submitted version, we evaluated FastV only on Video-LLaVA-Qwen because its official implementation provides pruning only for LLaMA-based language models. To make it work with Qwen2 and Qwen2.5 under FlashAttention 2, we had to replace the k-1 attention layer with an eager version to obtain attention maps, which added noticeable overhead (clarified in the revised version). To avoid inconsistent latency, we limited FastV to that backbone in the first submission. In the new experiments, we now report FastV at multiple retention ratios on both Qwen2.5-VL-7B and LLaVA-OneVision. Across these settings, FastV shows larger accuracy drops than SURGE at the same retention, especially at lower budgets.
>
> **Q2 Complementarity with AKS.** We appreciate the reviewer highlighting this. In the submitted version, our claim that AKS and SURGE are "complementary" was based on combining a frame-selection method (AKS or BOLT) with SURGE, which was somewhat too strong. At similar token budgets, AKS+SURGE often gives accuracy very close to using AKS or SURGE alone. This is also intuitive. Once AKS has already filtered out many redundant frames, the remaining content is less redundant, so an additional token-level pruning stage has limited room to improve accuracy at the same overall budget. In the revision, we will make clear that using AKS with SURGE is mainly to handle longer videos or tighter budgets, not to increase accuracy when the token budget stays the same.
>
> In the new experiments, we also identified a more direct form of complementarity: SURGE's surprise score can serve as an alternative importance signal inside existing token-merging methods, like VisionZip. VisionZip selects dominant tokens using vision-tower attention scores before merging the remaining tokens. In our experiments, we keep VisionZip's merging step but replace its attention-based dominant-token selection with SURGE's surprise scores. Across benchmarks and retention ratios, this "Surprise+VisionZip" variant is consistently comparable to, and in many cases slightly better than the original VisionZip at the same budget, with the clearest gains on LongVideoBench. This suggests that in many contexts the surprise signal is more stable than attention scores for identifying important tokens, and that SURGE integrates naturally with such token-merging pipelines, not only with frame-selection methods.
>
> **Q3 Results on Next-QA and EgoSchema.** Thank you for the suggestion. We have now included both datasets in the new experiments (see tables above). The trends are consistent with the other benchmarks: SURGE and SURGE* maintain accuracy very close to the full-token baseline across all retention ratios, while providing substantial token reduction. On LLaVA-OneVision, DyCoKe performs competitively in accuracy, but as discussed in our response to reviewer xZY7, SURGE achieves this quality with lower FLOPs, lower latency overhead, and without relying on internal attention maps from the language model, which makes it more adaptable across architectures.
>
> **Q4 Comparison with recent training-free methods.** We have now added comparisons with the latest methods in the tables above.
>
> **Q5 Typographical issue (L49 "surprise").** Thank you for pointing this out. The quotation marks were automatically formatted by the template, so we now use \textit{surprise} instead.

---

> ### Author Response · Authors · 2025-11-27
>
> We would like to kindly follow up on our rebuttal and again thank the reviewer for the insightful suggestions. In response, we have added broader comparisons with recent training-free methods (DyCoKe, LLaVA-Scissor, VisionZip, FastV across multiple retention ratios), new results on EgoSchema and Next-QA, clarified the AKS discussion and corrected the noted typo.
>
> We also refer the reviewer to our discussion with Reviewer XSco, where we ran additional 256-frame, ~20k-token baselines in VLMEvalKit. These results support the validity of our setup and the reported gains.
>
> We would be grateful if the reviewer could let us know if any further clarification is needed.

---

### Author Response · Authors · 2025-11-30
**Brief Summary for the Area Chair**

We thank the AC and the reviewers for their time. Below, we briefly summarize how the reviews and the rebuttal stand during the whole process.

**Contribution and novelty.**
The paper proposes **SURGE**, a **training-free, backbone-agnostic token reduction method** for long-video VLMs, based on a **prediction-error–driven "surprise" score in token space**. Several reviewers explicitly highlight that this formulation is **novel, conceptually elegant and theoretically well-grounded**, and that it offers a principled alternative to more ad-hoc attention- or CLIP-based heuristics while remaining practical to integrate into existing VLMs.

**Reviewer positions after rebuttal.**

* **Reviewer xZY7** started borderline but, after seeing the extended comparisons and FLOPs/latency analysis, **raised their score to 6** and notes that their main concerns are now addressed.
* **Reviewer XSco** was already at **6** and asked for a stronger 256-frame, large-context baseline. Once we ran that experiment and showed that SURGE can slightly improve over the full-token baseline while using fewer tokens, they wrote that this **resolves their remaining concern** and kept a positive assessment.
* **Reviewer ZWDa** finds the method **well-motivated, technically justified, clearly written and reproducible**, and explicitly mentioned they would be willing to raise their score if broader comparisons were added. The rebuttal now includes exactly those additional baselines and datasets.
* **Reviewer ky99** describes the work as a **solid study**, considers the theoretical derivation of surprise **sound**, and sees surprise as a plausible alternative notion of saliency. Their remaining reservations are mostly about how to position this idea among many recent saliency proxies and about possible future extensions (earlier-stage pruning, fully streaming), rather than about correctness of the current method.

**Concerns addressed.**
Across the rebuttal and follow-up, we have:

* Added comparisons to recent training-free methods (DyCoKe, LLaVA-Scissor, VisionZip, FastV) under their standard evaluation setup.
* Provided quantitative overhead measurements (latency and FLOPs), showing that SURGE sits in the same low-cost regime as the most efficient baselines.
* Run the 256-frame, large-context experiment requested by XSco, where SURGE matches or slightly improves the base model while using fewer tokens.
* Included hyperparameter sweeps to demonstrate robustness and clarified how SURGE can be used causally, with SURGE* explicitly framed as an offline, query-aware extension.
* Made textual and formatting updates, including clearer explanations of where surprise is computed, how streaming usage works, and an expanded related-work section, along with fixing minor layout issues.

Overall, the initial reviews already regarded the method as technically sound, practically relevant, and empirically well supported, and the discussion has mainly focused on experimental breadth, efficiency characterization, and positioning rather than on fundamental correctness. With the additional experiments and clarifications now in place, we hope this concise summary of the main concerns and our responses is useful context when you assess the submission.

---

### Meta-Review · Area_Chair_PEDb · 2025-12-22

**Summary:**

**Paper Summary**

The paper studies a training-free token compression method for video understanding models. The main compression is driven by the "surprise", which is mostly a temporal-smoothness clue. The compression is done in two stages, the first stage identify "surprise" tokens from the patch embedding, while the second stage identity key frames.

**Reasons of Recommendation**

The main concern of the papers are missing baselines and speed details. The rebuttal fixed it. There are no clear factual issues after rebuttal. Besides this, the paper's method has space to be improved, and it has been reflected in the score. Thus the decision can be made from the adjusted score.

**Review's concerns**

The main concerns of the papers are:
1. Lack of evaluations and benchmarks (ZWDa, xZY7, XSco, ky99)
--> `Mostly resolved during rebuttal`
2. The method with "constant-velocity prediction plus affine drift fitting" might be under-explored as video dynamics can be more complex. (xZY7, ky99).  --> `This question is not fully resolved; but it will be set as minor as every method is not perfect.`
3. The sensitivity to hyperparameters (xZY7)  --> `Mostly resolved during rebuttal`
4. The improvement of the method is not significant (xZY7, XSco). --> `A little bit. ICLR discourages SOTA game thus this point will also be minor`.
5. The speed is not quantified (XSco, ky99) --> `Provided in rebuttal`

**Note**
1. One reviewer highlighted the "format requirement" on table number/captions. We would not execute this rule and would give chance to authors for correction.

**Reviewer Concerns:**

Please see above.

**Reviewer Scores:**

xZY7 and ZWDa will increase to 6. xZY7 has committed in response while ZWDa's main concerns are answered.

The final score would be 6, 6, 6, 4.

---

### Decision · Program_Chairs · 2026-01-26

Accept (Poster)